# SPO: A Black-box, Unbiased, Robust Watermarking Method for Large Language Model

## Abstract

Large language models (LLMs) have revolutionary impacts on text generation. Despite their widespread application, LLMs raise significant ethical and security concerns about potential misuse, such as fake news and malicious content. Watermarking technology is known as a crucial means of distinguishing generated content and then mitigate misuse. Existing watermarking methods have their respective strengths and weaknesses, but it remains a challenge to achieve a balance between black-box embedding, unbiased output, and robustness. To address this limitation, we propose a novel black-box watermarking method called the Sampling and Prioritizing Output method (SPO). Through prioritizing the allocation of watermarked tokens over non-watermarked tokens, the SPO method maximizes the number of watermarked tokens within the designated watermarked subspace. Subsequently, the method randomly samples an output token from this subspace to effectively embed the watermark. As a black-box approach, the SPO method does not rely on detailed model parameters for watermark embedding and effectively safeguards intellectual copyrights of LLMs. Extensive experimental results and theoretical analysis indicate that the SPO method is an unbiased method that embeds the watermark without compromising the quality of generated content. Furthermore, it exhibits superior detectability and robustness compared to existing unbiased watermarking methods. This achievement addresses remarkable advantages over current unbiased methodologies, providing a possible solution that adapts better to real-world scenarios.

## 1 Introduction

In recent years, Large Language Models (LLMs) have advanced at an unprecedented pace (OpenAI, 2023; DeepSeek-AI, 2024). The context generated by these models can even match human-level quality, making it increasingly challenging to differentiate between machine-generated content and human-authored content. In addition, the easy accessibility and affordability of these models make it possible for irresponsible users to exploit them for malicious purposes. Issues such as the generation of fake news, the creation of malicious content, and even the fabrication of academic papers are closely linked to the misuse of LLM (Pan et al., 2023; Kim et al., 2024). It is crucial to curb the potential misuse of such technology by effectively distinguishing between the content generated by LLMs and that created by humans (Chakraborty et al., 2024; Mitchell et al., 2023) .

Watermarking technology is known as an effective solution to prevent the misuse of LLM (Kirchenbauer et al., 2023b; Li et al., 2024; Wu et al., 2023). To date, the main watermarking methods have their respective pros and cons. For example, the KGW method (Kirchenbauer et al., 2023a) leverages the bias characteristics of the red-green list, thus achieving robustness and high detectability. However, this kind of watermark often degrades the quality of the generated content, which is classified as a biased method. Subsequently, the unbiased watermarking method (Hu et al., 2024) is introduced, which selects watermarked tokens by random sampling and maintains the quality of the generated content. However, the robustness of these unbiased methods is relatively weak, potentially leading to the failure to extract the watermark under attacks. In addition, the researchers proposed the STA-M method (Mao et al., 2025), a black-box watermarking technique that operates independently of the details of LLMs such as the probability distribution, making it effective in scenarios where detailed model parameters are inaccessible. When hyperparameter M is set to 1, this method rejects non-watermarked tokens once and yields unbiasedness, but its robustness remains

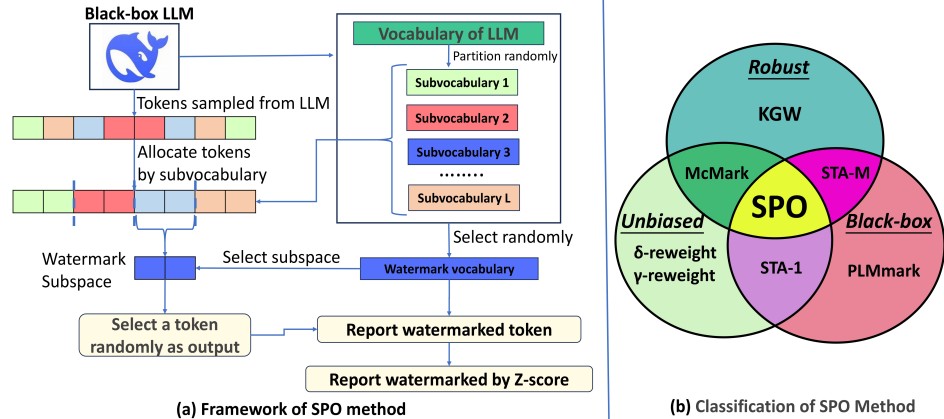

Figure 1: The overview of the SPO method. (a) introduces two sequential processes of SPO method: in process of embedding, sample multiple candidate tokens and allocate them to subspaces by partitioned subvocabularies. Then output token is randomly sampled from one subspace corresponding to watermark vocabulary. In process of detection, conduct a Z-test to confirm the watermark by calculating the number of watermarked tokens. (b) introduces the classification of watermarking techniques based on three criteria: black-box, unbiased, and robust.

insufficient. In contrast, when M is greater than 1, the method repeatedly rejects and gains improved robustness at the cost of losing its unbiased feature. As shown in Figure 1, despite achieving specific goals in particular scenarios, it is clear that existing watermarking methods still fail to achieve a balance between three criteria: black-box embedding, unbiased output and robustness, which are commonly required in practical use. Therefore, it is imperative to apply reasonable controls with a new watermarking method that balances between these criteria, then satisfies the requirements of practical circumstances.

To address this challenge, we propose an innovative solution called the Sampling and Prioritizing Output (SPO) method. As shown in Figure 1, this method obtains multiple candidate tokens from the LLM and allocates them to corresponding subspaces according to randomly divided subvocabularies. Then, a watermark subvocabulary is selected at random to choose the watermark subspace, from which tokens are sampled as output. We prioritize the allocation of watermarked tokens to watermark subspace over the non-watermarked tokens to maximize the number of watermarked tokens and construct an extra allocation of superfluous tokens to achieve the unbiased output. In addition, the embedding of the watermark intrinsically does not rely on the probability of LLMs, meaning that it is a black-box watermarking method and can effectively safeguard the copyright of model owners. During the detection phase, the subvocabularies are reconstructed with the watermark key, and the presence of the watermark is determined by checking the number of tokens included in the watermark vocabularies.

Our main contributions are summarized as follows.

1. We propose the Sampling and Prioritizing Output (SPO) method, which achieves novel watermark embedding by maximizing the number of watermarked tokens in subspaces. The embedding is entirely independent of LLMs' specific parameters (e.g. probability distributions), making it a purely black-box watermarking method.

2. The SPO method is an unbiased watermarking technique that maintains the quality of generated content while also exhibiting strong robustness and excellent detectability, comparable to other unbiased methods.

3. The SPO method is characterized by achieving balance between black-box embedding, unbiased output, and robust performance, which remain unsolved by current methods. We hope that this method can significantly promote the application of watermarking techniques in LLM and address the challenges of complex real-world scenarios.

## 2 RELATED WORK

### 2.1 BIASED WATERMARK

In recent years, many large language model watermarking algorithms have been used for generated text detection (Kamaruddin et al., 2018; Yoo et al., 2024; Yang et al., 2022). As the first LLM watermark algorithm, the KGW method (Kirchenbauer et al., 2023a) randomly divides the vocabulary into a red list and a green list, and embeds the watermark by encouraging token generation in the green list. To reduce the impact on the quality of text generation, the SWEET method (Lee et al., 2024) uses the entropy of the text to adaptively modify logits. To strengthen the robustness of the watermark, the UNIGRAM method (Zhao et al., 2024) fixes the division of the red and green list then embeds the watermark.

### 2.2 UNBIASED WATERMARK

To avoid damaging the quality of text generation, unbiased watermark algorithms such as $\delta$-reweigt and $\gamma$-reweight (Hu et al., 2024) and DiPmark (Wu et al., 2024) are proposed to maintain the excepted probability distribution, then achieve unbisaed output. These methods use a reweight algorithm that randomly resets the logits to increase the probability of outputting watermarked tokens to embed the watermark.

### 2.3 BLACK-BOX WATERMARK

The black-box watermarking method does not rely on specific model parameters during watermark embedding. The PLMmark method (Li et al., 2023) establishes a strong link between a digital signature and trigger words to embed the watermark. The SynthID-Text method (Dathathri et al., 2024) randomly assigns weights to the sampled tokens and generates watermarked tokens using tournament sampling efficiently and unbiasedly. The STA-M method (Mao et al., 2025) uses rejection sampling to randomly divide the vocabulary into green and red lists, refusing to output tokens from the red list, and resampling to increase the probability of outputting tokens from the green list, thus embedding the watermark.

## 3 METHODOLOGY

**Definition** (Subvocabulary and Subspace) Given the LLM vocabulary $V$, we divide the vocabulary into uniform parts, each part being defined as a subvocabulary. For example, $V[i]$ denotes the $i$-th part of vocabulary, referred to subvocabulary $i$. Each subvocabulary has a specific list to storge the tokens sampled from LLM. We define this list as a corresponding subspace for the subvocabulary and abbreviate it as "subspace(s)" when referring to multiple subvocabularies.

### 3.1 VOCABULARY-DIVISION STRATEGY

The prior methods (Kirchenbauer et al., 2023a; Mao et al., 2025) often embed the watermark by dividing the vocabulary into two parts and choosing one as the watermark vocabulary. Meanwhile, researchers (Chen et al., 2025) point out that increasing the number of uniform subvocabularies within a specific range can substantially improve watermark performance in white-box environments. Inspired by this result, we adopt the strategy of dividing the LLM vocabulary into multiple uniform subvocabularies and randomly selecting one as the watermark vocabulary. Then we develop our novel Sampling and Prioritizing Output (SPO) method.

### 3.2 EMBEDDING BY THE SPO METHOD

As a black-box approach, the SPO method extracts several tokens as candidate tokens from which the output token is chosen rather than reweights the logits of LLM like white-box methods. To control the progress of embedding, we introduce two hyperparameters: L, which represents the number of subvocabularies after partitioning, and N, which denotes the number of tokens sampled from the LLM before generating the current token. We build L subspaces to classify tokens by their

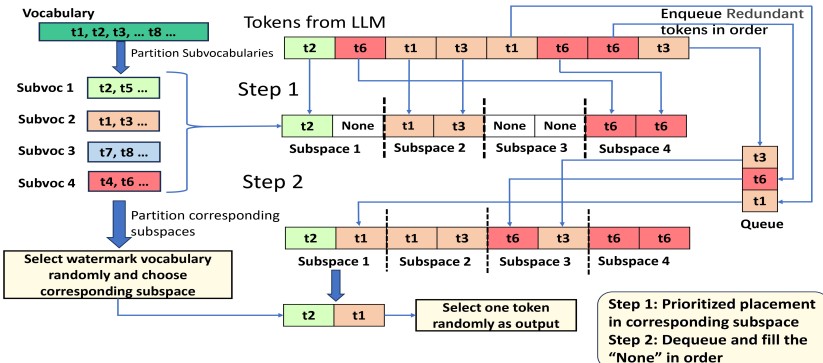

Figure 2: An example of generating the token by SPO method. Sample 8 candidate tokens and partition the vocabulary randomly. Then, 4 uniform subspaces are created, which contains 2 tokens calculated by N/L. Step 1: The candidate tokens are sequentially allocated to their respective subspaces. If a subspace is full, the token is placed in a queue. Step 2: tokens are dequeued and placed into empty positions until all subspaces are filled. Finally, a subvocabulary is randomly selected and sample a token randomly from the corresponding subspace as output.

subvocabularies, and each subspace contains N/L tokens to ensure that every token can be properly allocated.

Provided that 8 tokens are sampled and the vocabulary is partitioned into 4 subvocabularies. Ideally, the 8 sampled tokens originate evenly from each of the 4 subvocabularies. In this scenario, tokens are sequentially allocated to their corresponding subspaces and each subspace is filled. Then, a vocabulary is randomly selected as the watermark vocabulary, and the corresponding subspace is chosen for output. This kind of allocation guarantees that the watermarked tokens within the selected subspace can reach their maximum, then successfully embed the watermark. Not only does it maintain an unbiased nature but also achieves the highest success rate of watermark embedding. However, this ideal scenario faces a challenge in practical applications. As illustrated in Step 1 of Figure 2, discrepancies arise where the number of tokens in certain subvocabularies exceeds the capacity of their respective subspaces; additionally, other subspaces contain fewer tokens than they are able to accommodate. This issue introduces mismatches and bias when these underfilled subspaces are selected for output.

To solve this problem, we design a pioneering solution: employing an extra queue to store redundant tokens if a subspace is filled. As shown in Step 2 of Figure 2, after Step 1, if there are empty positions "None" in any subspace, tokens in the queue are sequentially dequeued and placed in these positions until the queue becomes empty and all subspaces are filled. Subsequently, a subvocabulary is randomly selected as the watermark vocabulary, and its corresponding subspace is designated as the watermark space. Within this subspace, a token is randomly chosen for the final output, which completes the embedding process of the SPO watermark. This method can retain unbiased characteristics (as proven in Appendix A) and maximize the probability of outputting watermarked tokens by prioritizing the placement of tokens from each subvocabulary in their corresponding subspaces. As a result, this enhancement in probability elevates the detectability of the watermark.

### 3.3 DETECTION BY SPO WATERMARK

The SPO method involves sampling and prioritizing output strategy, which ensures that the number of watermarked tokens in the corresponding subspace is maximized and increases the likelihood of outputting tokens from the watermark vocabulary, thus effectively embedding the watermark. In particular, the partitioning and selection of subvocabularies depend on information like the watermark key, rather than on specific model parameters. Consequently, the extraction of SPO watermark can be performed precisely in a black-box environment.

In watermark detection, we reconstruct the watermark vocabulary using the watermark key to determine whether the current token belongs to the watermark vocabulary. If the token is identified

---

**Algorithm 1** SPO Embedding Method

---

**Input**: LLM $M$, prompt $p$, hyperparameter $N$ and $L$, watermark key $k$, vocabulary $V$
**Output**: generated token $x$

    Obtain token list $T$ containing $N$ candidate tokens from LLM $M$ according to prompt $p$
    Partition L subvocabularies from vocabulary $V$
    Construct subspace list $S$ with $L$ subspaces. Each subspace contains N/L positions
    **for** $j = 0, 1, \ldots, N - 1$ **do**
        Get the index $i$ of token $T[j]$ according to subvocabularies
        **if** $S[i]$ has empty position **then**
            Put $T[j]$ in $S[i]$ sequently
        **else**
            Put $T[j]$ in queue
        **end if**
    **end for**
    **for** $j = 0, 1, \ldots, L - 1$ **do**
        **if** $S[j]$ has empty position **then**
            Dequeue and put in empty position of $S[j]$ subsequently
        **end if**
    **end for**
    Select subvocabulary $V[w]$ according to $k$ and sample output token $x$ randomly from corresponding subspace $S[w]$
    **return** $x$

---

as part of the watermark vocabulary, it is reported as a watermarked token. After traversing the entire text, construct a Z-test to confirm the watermark. Assuming that the number of tokens in the detected text is $n$, we build our detection of the watermark on the following null hypothesis ($H_0$): In the absence of the watermark, the probability that the LLM selects tokens from each subvocabulary is uniform, $\frac{1}{L}$ for each subvocabulary. Therefore, under the null hypothesis, the probability that a token belongs to the watermark vocabulary is $\frac{1}{L}$, and the probability that it does not belong is $1 - \frac{1}{L}$. As a result, the number of watermarked tokens $W$ follows a Binomial distribution $B(n, \frac{1}{L})$:

$$Z = \frac{W - n \cdot \left(\frac{1}{L}\right)}{\sqrt{n \cdot \left(\frac{1}{L}\right) \cdot \left(1 - \frac{1}{L}\right)}} \tag{1}$$

where $W$ represents the number of watermarked tokens in the detected text. If the Z-score exceeds the threshold, the text contains a sufficient number of watermarked tokens, and generating such a text without knowledge of the watermarking rule is a rare event. This means that the null hypothesis is rejected, and the text carries a watermark. Conversely, if the null hypothesis is not rejected, the text does not carry a watermark.

We set the threshold of the Z-test as $Z_\alpha$, and the threshold of $W$ as $W_\alpha$. The theoretical false positive rate is calculated as follows:

$$P(Z > Z_\alpha) = \sum_{k=W_\alpha}^{n} \binom{n}{k} \left(\frac{1}{L}\right)^k \left(1 - \frac{1}{L}\right)^{n-k} \tag{2}$$

where $W_\alpha = \lceil Z_\alpha \cdot \left(\frac{1}{L}\right) \cdot \left(1 - \frac{1}{L}\right) + N \cdot \left(\frac{1}{L}\right) \rceil$

### 3.4 THEORETICAL ANALYSIS

The SPO method achieves the embedding of the watermark by a clever embedding strategy. However, similar to other unbiased methods, situations may arise where the token sampled from watermark subspace does not belong to the watermark subvocabulary and is not detected as a watermarked token. Following the analytical framework of existing studies (Chen et al., 2025), we define this scenario as the "True Negative" (TN), and use the corresponding metric, the "True Negative Rate"

(TNR), to reflect the detectability of the watermark. Specifically, a lower TNR indicates a greater number of valid watermarked tokens, thereby enhancing the detectability of the watermark.

To further elaborate, assume that the watermark vocabulary is denoted as $w$, and its corresponding subspace is $S_w$, which contains $\frac{N}{L}$ tokens. The occurrence of a True Negative falls into two scenarios:

1. All tokens within the subspace $S_w$ are completely unrelated to the watermark vocabulary $w$. In this case, no valid watermarked token can be generated regardless of the sampling approach.

2. Part of the tokens within the subspace $S_w$ belong to the watermark vocabulary $w$, with $i$ tokens in total of $\frac{N}{L}$. In this situation, the probability of randomly sampling a valid watermarked token is $\frac{i}{\frac{N}{L}}$.

The sum of the probabilities of these two scenarios represents the theoretical True Negative Rate. Suppose that the probability of an output token belonging to the watermark vocabulary is $P_w$ (where $0 \leq P_w \leq 1$), then the theoretical formula for the True Negative Rate (TNR) can be expressed as:

$$E_{TNR} = E(\text{scenario 1}) + E(\text{scenario 2}) =$$
$$(1 - P_w)^N + \sum_{i=1}^{N/L} \left( \frac{N/L - i}{N/L} \right) \binom{N}{i} (P_w)^i (1 - P_w)^{N-i} \tag{3}$$

Considering two hyperparameters: $N$ and $L$, we selectively calculate the excepted true negative rate under specific conditions and compare it with the result of other methods. Assuming a uniform probability distribution of the LLM, $P_w \in [0, 1]$. To investigate the impact of hyperparameters, we set the number of subvocabularies $L$ to $\{2, 3, 5, 10, 20\}$ and correspondingly set the subspace size $N/L$ to $\{1, 2, 3, 4, 5, 6\}$.

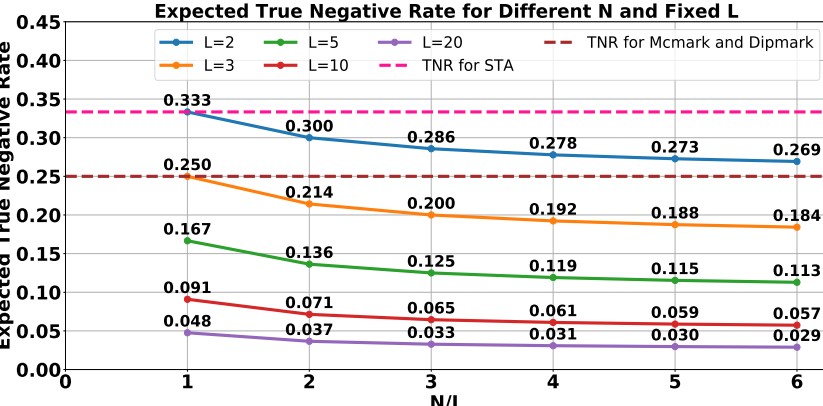

Figure 3: Impact of exponentially increasing $N$, on the expected true negative rate under fixed number of $L=\{2,3,5,10,20\}$, which represents the number of partitioned subvocabularies. The vertical axis represents the predicted true negative rate (TNR), while the horizontal axis indicates the subspace size $N/L$.

As shown in 3, when $N=4$ and $L=2$, the true negative rate becomes $\frac{3}{10}$. Furthermore, setting $N=3$ and $L=3$, as well as $N=5$ and $L=5$, results in true negative rates of $\frac{1}{4}$ and $\frac{1}{6}$, respectively. In addition, we select three representative watermarking methods for comparison: STA-1 (Mao et al., 2025), MCmark (Chen et al., 2025) and DiPMark (Wu et al., 2024). According to existing studies, the true negative rates for the STA-1, Mcmark, and DiPMark methods are $\frac{1}{3}$, $\frac{1}{4}$ and $\frac{1}{4}$, respectively. Clearly, as $N$ and $L$ increase, the SPO method reveals a true negative rate lower than that of other comparison methods (specifically $\frac{1}{4}$ and $\frac{1}{3}$). This implies that the SPO method has a smaller probability of failure

in embedding and can detect more valid watermarked tokens during the detection process, ultimately improving the detectability of the watermark.

So we can conclude that increasing both $N$ and $L$ within a specific range leads to a reduction of the expected true negative rate, thus enhancing the detectability of the watermark. Detailed analysis is provided in the appendixA. The subsequent analysis further confirms the superior performance of our method.

## 4 EXPERIMENTS

In order to demonstrate the advantages of the SPO method, a series of experiments are carried out mainly on four key aspects: unbiasedness, robustness, detectability, and applicability study. Furthermore, we compare the experimental results with those of typical watermarking methods to highlight the superior and comprehensive performance of the SPO approach. Detailed descriptions of the experimental configuration and the selection of comparison methods are provided in Appendix C.

### 4.1 UNBIASEDNESS

To verify the unbiasedness of the SPO method, we create multiple configurations with different parameters $N$ and $L$ and carry out comparative analyzes with both unbiased and biased approaches. Using current experimental conditions (Hu et al., 2024), the results are presented in Table 1. In both machine translation and text summarization tasks, the generated content with SPO watermark shows high consistency in machine evaluation metrics compared to the non-watermarked counterpart, with only minor fluctuations resulting from variations in output length and diversity. The performance of the SPO method does not show significant differences compared to other unbiased mainstream methods such as $\delta/\gamma$-reweight (Hu et al., 2024) and DiPmark (Wu et al., 2024). Furthermore, in comparative experiments, the biased watermarking method, such as KGW (Kirchenbauer et al., 2023a), exhibits a significant decrease in generation quality when the bias parameter $\delta$ increases, while the SPO method still maintains stable generation quality in multiple parameter configurations of $N/L$. This indicates that the SPO watermarking method is unbiased, and the quality of watermarked text remains unaffected in practical applications, thus maintaining normal functionality for users.

Table 1: Performance of different methods on TS and MT. We amplify BERTScore and ROUGE with a factor of 100

| Type | Method | Machine Translation | | Text Summarization | | |
|---|---|---|---|---|---|---|
| | | BERTScore | BLEU | BERTScore | ROUGE-1 | Perplexity |
| Baseline | No watermark | 56.2 | 21.7 | 32.67 | 38.65 | 5.031 |
| Biased | KGW ($\delta$=1.0) | 55.6 | 21.3 | 32.32 | 38.20 | 5.229 |
| | KGW ($\delta$=2.0) | 54.1 | 19.5 | 31.21 | 37.14 | 6.252 |
| Unbiased | $\delta$-reweight | 56.1 | 21.7 | 32.71 | 38.60 | 5.008 |
| | $\gamma$-reweight | 56.3 | 21.8 | 32.77 | 38.70 | 5.007 |
| | DiPmark ($\alpha$=0.3) | 56.2 | 21.9 | 32.73 | 38.64 | 5.025 |
| | SPO (N=2,L=2) | 56.5 | 22.2 | 32.78 | 38.64 | 5.080 |
| | SPO (N=4,L=2) | 56.3 | 22.1 | 32.64 | 38.42 | 5.123 |
| | SPO (N=8,L=2) | 56.1 | 21.7 | 32.62 | 38.39 | 5.096 |
| | SPO (N=20,L=20) | 56.7 | 22.3 | 32.37 | 38.15 | 5.118 |

### 4.2 DETECTABILITY

We systematically evaluate the detectability of the SPO method based on the C4 (Raffel et al., 2020) dataset by analyzing the true positive rate (TPR). As shown in Table 2, we compare the TPR of the SPO method with other watermarking methods under fixed FPR settings. The experimental results indicate that the SPO method exhibits outstanding detectability. When the parameters are

set to $N = 4$ and $L = 2$, the detection performance of the SPO method is comparable to that of unbiased watermarking methods. Furthermore, when the parameters are changed to $N = 20$ and $L = 20$, the result outperforms all existing unbiased watermarking methods, even surpassing the biased KGW method (Kirchenbauer et al., 2023a). The STA-M method demonstrates superior detection capabilities and significantly lower statistical p-values; however, these advantages come at the cost of compromised unbiasedness and potential risks to robustness. In contrast, the SPO method achieves dynamic optimization of detection performance through flexible parameter tuning, preserving the essential unbiased nature and thus better meeting practical application requirements. (see Appendix C for additional experiments).

Table 2: TPR of generated-text detection for different methods under fixed FPR when max-length of generation is 100

| Type | Method | TPR@FPR=0.1 | TPR@FPR=0.05 | TPR@FPR=0.01 | Average $p$-value |
|---|---|---|---|---|---|
| Biased | KGW ($\delta$=1) | 0.847 | 0.799 | 0.585 | $7.25 \times 10^{-2}$ |
| | KGW ($\delta$=2) | 0.956 | 0.938 | 0.868 | $2.24 \times 10^{-2}$ |
| | STA-8 | **0.998** | **0.998** | **0.996** | $3.92 \times 10^{-4}$ |
| | STA-16 | **1.000** | **1.000** | **1.000** | $6.90 \times 10^{-7}$ |
| Unbiased | STA-1 | 0.982 | 0.970 | 0.878 | $5.60 \times 10^{-3}$ |
| | $\delta$-reweight | 0.853 | 0.820 | 0.716 | $2.54 \times 10^{-1}$ |
| | $\gamma$-reweight | 0.937 | 0.918 | 0.868 | $1.04 \times 10^{-1}$ |
| | DiPmark ($\alpha$=0.3) | 0.871 | 0.838 | 0.866 | $4.47 \times 10^{-2}$ |
| | SPO (N=4,L=2) | 0.910 | 0.853 | 0.754 | $2.69 \times 10^{-2}$ |
| | SPO (N=20,L=20) | **0.986** | **0.978** | **0.966** | $7.80 \times 10^{-3}$ |

## 4.3 ROBUSTNESS

Based on the C4 dataset (Raffel et al., 2020), we perform an analysis using common replacement attacks to thoroughly evaluate the robustness of the SPO method. The AUC results of the SPO method under varying attack strengths are presented in Table 3. In a malicious attack, the $\delta$-reweight (Hu et al., 2024) method exhibits a substantial decrease in AUC, which compromises the integrity of watermark extraction. In contrast, the SPO method demonstrates the least degradation in AUC, with its watermark extraction capability remaining largely intact. This behavior validates the superior robustness of the SPO method against attacks (see Appendix C for more experiments of robustness with different attacks). The advantage of robustness comes from the ability of the SPO method to maximize the probability of outputting watermarked tokens, generating as many watermarked tokens as possible. Consequently, erasing the watermark requires more extensive modifications to reduce the $p$-value below the threshold level. When faced with random modifications, the SPO method detects the watermark more effectively, showcasing its improved robustness. Although increasing the hyperparameter $L$ (the number of subvocabularies) may have a negative impact on robustness (detailed analysis provided in Appendix C), the SPO method still outperforms other unbiased watermarking methods in the experimental setting of $N = 20$ and $L = 20$, demonstrating its strong robustness.

Table 3: AUC of watermark detection under different perturbation strength of replacement attack and max-length of generation

| Method | strength=0.0 | | | strength=0.05 | | | strength=0.1 | | | strength=0.2 | | |
|---|---|---|---|---|---|---|---|---|---|---|---|---|
| | 50 | 100 | 200 | 50 | 100 | 200 | 50 | 100 | 200 | 50 | 100 | 200 |
| KGW ($\delta$=1) | 0.914 | 0.945 | 0.946 | 0.904 | 0.935 | 0.937 | 0.883 | 0.920 | 0.929 | 0.849 | 0.896 | 0.901 |
| KGW ($\delta$=2) | 0.980 | 0.983 | 0.983 | 0.975 | 0.980 | 0.983 | 0.969 | 0.975 | 0.981 | 0.942 | 0.960 | 0.972 |
| STA-1 | 0.991 | 0.991 | 0.989 | 0.989 | 0.988 | 0.984 | 0.984 | 0.984 | 0.980 | 0.971 | 0.965 | 0.967 |
| $\delta$-reweight | 0.864 | 0.916 | 0.932 | 0.796 | 0.841 | 0.887 | 0.686 | 0.750 | 0.776 | 0.592 | 0.673 | 0.636 |
| $\gamma$-reweight | 0.966 | 0.971 | 0.981 | 0.947 | 0.959 | 0.975 | 0.863 | 0.918 | 0.945 | 0.711 | 0.744 | 0.829 |
| DiPmark ($\alpha$=0.3) | 0.946 | 0.956 | 0.962 | 0.905 | 0.927 | 0.946 | 0.828 | 0.873 | 0.909 | 0.686 | 0.738 | 0.785 |
| SPO (N=20,L=20) | **0.996** | **0.992** | **0.993** | **0.994** | **0.992** | **0.993** | **0.993** | **0.991** | **0.992** | **0.989** | **0.989** | **0.989** |

## 4.4 BALANCE BETWEEN THREE CRITERIA

The experimental results demonstrate that current watermarking methods exhibit strengths in evaluation. Specifically, the KGW (Kirchenbauer et al., 2023a) method demonstrates superior robustness and detectability; however, it lacks the ability to achieve black-box embedding and unbiased output. The DiPmark (Wu et al., 2024) method successfully achieves unbiased output, yet its embedding process depends on sensitive parameters such as logits. The STA-1 (Mao et al., 2025) method achieves black-box embedding and unbiased embedding; however, its robustness remains inferior compared to that of KGW. In contrast, the STA-M (Mao et al., 2025) method effectively improves robustness but sacrifices the advantage of unbiased output. Notably, existing methods have yet to achieve a balance between black-box embedding, unbiased output and robustness. In comparison, the SPO method not only achieves black-box embedding but also surpasses the biased KGW method in terms of robustness. Although its detectability is slightly inferior to that of the black-box method STA-M, the SPO method maintains the characteristic of unbiased output, thus effectively balancing three significant criteria, and better meeting practical requirements.

## 4.5 APPLICABILITY STUDY

Based on high and low entropy datasets, combined with two models, we systematically evaluate the detectability of the SPO method across different scenarios. As illustrated in Figure 4, the experimental results demonstrate not only the excellent performance of the SPO method in various environments but also provide a comparison result with the mainstream watermarking methods (Hu et al., 2024; Wu et al., 2024; Kirchenbauer et al., 2023a; Mao et al., 2025). This outcome highlights the excellent adaptability of the method to diverse environments. In particular, like other methods, the detectability of the watermark is slightly influenced by the model or dataset. However, the overall performance fluctuation of the SPO method remains minimal, ensuring its stability and reliability (see Appendix C for additional analysis of applicability).

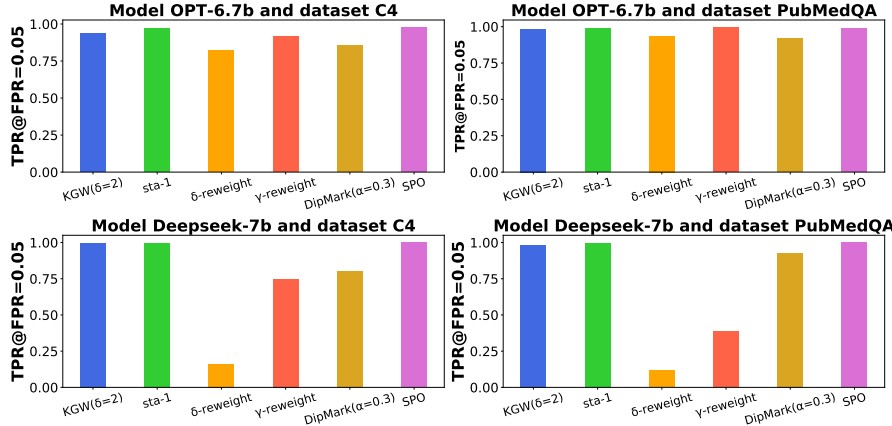

Figure 4: Applicability study of SPO method. We choose OPT-6.7b and Deepseek-llm-7b-base as ouptput model, then report results of TPR@FPR=0.05 on two datasets: C4 and PubMedQA.

## 5 CONCLUSION

We propose a novel watermarking method named SPO, which maximizes the probability of outputting watermarked tokens through sampling and prioritizing output strategy, thus achieving efficient watermark embedding. As a black-box watermarking technique, this method embeds watermarks without detailed LLM parameters, providing enhanced protection for LLM owners. Experiments demonstrate that the SPO method is unbiased across various tasks and has consistent performance in detectability and robustness under different conditions. We hope that this method can be deeply integrated into LLMs, providing innovative solutions for copyright protection and misuse prevention, thereby advancing the secure and widespread adoption of LLMs.

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

## A APPENDIX OF THEORETICAL ANALYSIS

### A.1 ANALYSIS OF EXPECTED TRUE NEGATIVE RATE

As shown in Figure 3, when the number of candidate tokens $N$ unchanged, increasing the $L$ leads to a decrease in the expected true negative rate, which is reflected in the overall downward change in the curve. For example, when the subspace size $N/L$ is 1, the true negative rate corresponding to $N$=2 is $\frac{1}{3}$, while for $N$=10, the true negative rate decreases to 0.091. When $N$=20, the true negative rate decreases further to 0.048. This result indicates that increasing $L$ significantly enhances detectability under fixed $N$.

Furthermore, when the number of subvocabularies $L$ is fixed, increasing the $N$ also results in a decrease in the true negative rate. The curve illustrates that as $N$ increases, the expected true negative rate consistently decreases, a trend observed in the five experimental sets. This feature aligns with the advantage of black-box embedding methods over white-box methods: black-box methods can sample multiple tokens at one time, whereas white-box methods typically sample a single token and use it as output. For example, the STA-M (Mao et al., 2025) method can obtain multiple candidate tokens by rejecting the sampled token to select the final watermarked token, while Synthid-text (Dathathri et al., 2024) uses random sampling on multiple tokens to determine the output result. We further optimize this process in the SPO method by sampling multiple candidate tokens and categorizing them into corresponding subspaces based on their associated subvocabulary, thus achieving prioritization output of watermarked tokens. This mechanism demonstrates the theoretical superiority of the SPO method and improves the detectability of the watermark.

In summary, within a certain range, increasing the value of hyperparameter $L$ reduces the expected false positive rate. Similarly, when $L$ is fixed, raising the value of $N$ also reduces the expected false positive rate, thus improving the detectability of the watermark. However, excessive increases in $N$ without careful consideration will incur additional computational overhead and decrease the efficiency of generation. Besides, elevating $L$ can undermine the robustness of the watermark. A comprehensive analysis of these trade-offs is presented in the following sections.

### A.2 THEORETICAL ANALYSIS OF UNBIASEDNESS

Inspired by McMark (Chen et al., 2025), Synthid-text (Dathathri et al., 2024), and STA (Mao et al., 2025), we creatively propose the SPO method. It maintains excellent robustness and detectability while still being unbiased; additionally, the embedding and extraction processes are entirely kept in a black-box environment. This represents the primary advantage of the SPO method: it can simultaneously achieve black-box embedding, unbiased output, and robust performance. We also theoretically analyze how the SPO method achieves the key metric of unbiasedness, which serves as supplement to our unbiased evaluation experiments.

According to existing study (Hu et al., 2024), an unbiased watermark method requires that the expected probability of token after embedding the watermark must be equal to the original probability distribution, that is, given context $x_{0:i-1}$, prompt $p$ and key $k$, for any generation step $0 < i <= n$, if the watermarked model $P_{M,w}$ satisfies the following equation:

$$P_M(x_i \mid x_{0:i-1}) = \mathbb{E}_{\theta_i \sim P_\Theta}\left[P_{M,w}\left(x_i \mid \boldsymbol{x}_{0:i-1}, \theta_i\right), p\right] \tag{4}$$

Then the watermarking method is unbiased.

Let the vocabulary be denoted as $\mathcal{V}$, divided into $L$ subvocabularies, marked sequentially. The $m$-th subvocabulary can be denoted as $\mathcal{V}[m]$, with a corresponding probability $p_m$ $(p_m \in [0,1])$. As shown in Figure 5, during the embedding process, the following three scenes may occur:

Scene 1: The watermark subspace is filled with watermarked tokens.

Scene 2: The watermark subspace is partially filled with watermarked tokens, with a quantity of $i$ $(0 < i < N/L)$.

Scene 3: The watermark subspace is filled with non-watermarked tokens.

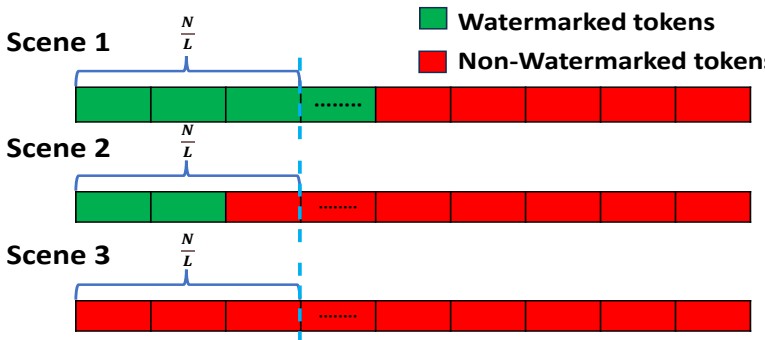

Figure 5: Three scenes of token distribution. To simplify the process, we set the watermark subspace to be left-aligned and use a blue dashed line to annotate the watermark subspace, meaning all areas to the left of the dashed line are used as the watermark subspace. Beside, we choose green to represent watermarked tokens and red to represent non-watermarked tokens.

Let token $j$ represent a randomly selected token from the vocabulary, with a corresponding probability $p_j$ ($p_j \in [0, 1]$). Considering the randomness of vocabulary partitioning, the probability that $j$ belongs to any subvocabulary is uniformly $1/L$.

To validate the unbiasedness of the method, we first consider a special case where $N = 2$ and $L = 2$. In this setting, two tokens are sampled and divided into two subvocabularies $v_1$ and $v_2$, with corresponding probabilities $p_1$ and $p_2$. Each subspace has a capacity of 1, which means that the probability of Scene 2 is 0, so we only take Scene 1 and Scene 3 into account for the analysis.

When token $j$ belongs to the watermark vocabulary, Scene 1 arises and there are three possible cases as illustrated in Figure 6:

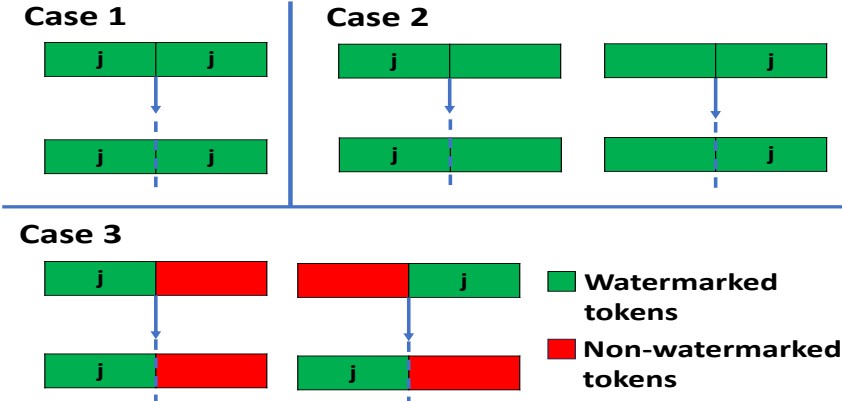

Figure 6: Assume that $N$=2,$L$=2. When token $j$ belongs to the watermark vocabulary, there are three cases. Note that in Case 2, the position of token $j$ influences the output result. However, under the conditions of Case 3, since $j$ is a watermarked token, it can be prioritized in the watermark subspace regardless of the order in candidate list, making the output result independent of the sequence.

Case 1: Both candidate tokens are $j$, resulting in probability of outputting $j$ is 1.

Case 2: One token is $j$, and the other is a different watermarked token (not $j$), resulting in the probability of outputting $j$ is $\frac{1}{2}$.

The SPO method adopts a sequential traversal approach; If the token $j$ is at the front of the sequence, it will be included in the watermark subspace and output. In contrast, if the token $j$ is in the back, it will not be allocated to the watermark subspace and output. When two tokens are sampled, the

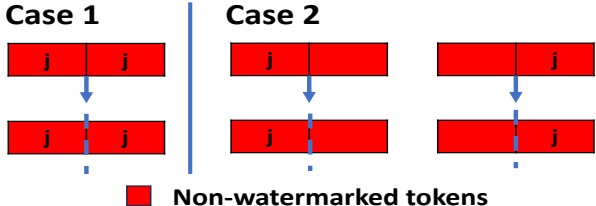

Figure 7: Assume that $N$=2,$L$=2. When token $j$ does not belong to the watermark vocabulary, there are two cases. Note that in Case 2, the position of token $j$ influences the output result.

probability that token $j$ is in the front or back is equally, and each probability is $\frac{1}{2}$. So, the the probability of outputting $j$ is $\frac{1}{2}$ in this case.

Case 3: One token is $j$, and the other is a non-watermarked token, resulting in the probability of outputting $j$ being 1.

Under the condition of Case 3, since $j$ is a watermarked token, it will be prioritized in the watermark subspace regardless of the order. So, the output of outputting token $j$ is independent of the sequence.

When token $j$ belongs to the watermark vocabulary, Scene 3 appears and the probability of outputting $j$ is 0.

According to the analysis, when token $j$ belongs to the watermark vocabulary, the total probability of outputting $j$ is calculated as:

$$\binom{2}{2} \cdot p_j^2 + \frac{1}{2} \cdot \binom{2}{1} \cdot p_j \cdot (p_w - p_j) + \binom{2}{2} \cdot p_j \cdot (1 - p_w) = p_j \cdot (2 - p_w) \tag{5}$$

Then we calculate the probability of outputting $j$ when token $j$ does not belong to the watermark vocabulary. In this situation, Scene 1 and Scene 3 arise, respectively.

For Scene 1, the probability of outputting $j$ is 0.

For Scene 3, there are two possible cases, as shown in Figure 7:

Case 1: Both candidate tokens are $j$, the probability of outputting $j$ is 1.

Case 2: One token is $j$, and the other is a non-watermarked token (not $j$), the probability of outputting $j$ is $\frac{1}{2}$.

Hence, when the token $j$ does not belong to the watermark vocabulary, the total probability of outputting $j$ is calculated as:

$$\binom{2}{2} \cdot p_j^2 + \binom{2}{1} \cdot \frac{1}{2} \cdot p_j \cdot (1 - p_w - p_j) = p_j \cdot (1 - p_w) \tag{6}$$

When embedding the watermark, the expected probability of outputting token $j$ is :

$$\frac{1}{2} \cdot p_j (2 - p_w) + \frac{1}{2} \cdot p_j (1 - p_w) = \frac{1}{2} \cdot p_j \cdot (3 - 2p_w) \tag{7}$$

Considering that the watermark vocabulary is randomly selected, with equal probability of choosing $v_1$ or $v_2$ as the watermark vocabulary and $p_1 + p_2 = 1$, the overall probability of outputting a watermarked token is: $p_w = \frac{1}{2}p_1 + \frac{1}{2}p_2 = 0.5$. Thus, the expected probability of outputting token $j$ is:

$$\frac{1}{2} \cdot p_j \cdot (3 - 2 \cdot 0.5) = p_j$$

It is evident that the expected probability of outputting token $j$ after embedding the watermark is consistent with that without embedding (both $p_j$). This result implies that the SPO method preserves

the output probability of any token $j$ as expected after watermark embedding, further verifying the unbiased nature of this method when $N = 2$ and $L = 2$.

Now, let us extend this study to verify that our method remains unbiased when $N$ and $L$ are arbitrary integers and $\frac{N}{L}$ is a positive integer.

Assuming that $\frac{N}{L} = 1$, we set the number of tokens $j$ in the candidate tokens as $i$ $(0 \leq i \leq N)$ and the number of watermarked tokens as $m$ $(0 \leq m \leq N)$. Similarly, each subspace has a capacity of 1, which means that the probability of Scene 2 is 0, so we simply use Scene 1 and 3 for analysis.

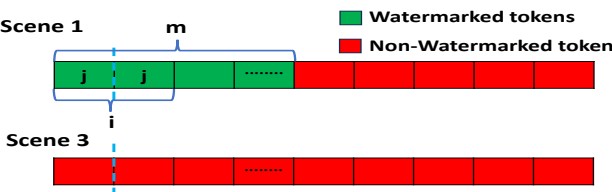

Figure 8: Assume that $\frac{N}{L} = 1$. When token $j$ belongs to the watermark vocabulary, there are two scenes of token distribution.

In Figure 8, we illustrate these two cases in detail. When token $j$ belongs to the watermark vocabulary, Scene 3 appears, $m = 0$ and the probability of outputting $j$ is 0.

When token $j$ belongs to the watermark vocabulary and Scene 1 appears, $m \geq 1$, the watermark subspace is randomly filled with watermarked tokens. The probability of selecting and outputting $j$ from $m$ watermarked tokens is $\frac{i}{m}$. The probability of outputting $j$ is given by:

$$\sum_{m=1}^{N} \left( \frac{i}{m} \cdot \left[ \binom{N}{m} \cdot p_w^m \cdot (1 - p_w)^{N-m} \right] \cdot \sum_{i=1}^{m} \binom{m}{i} \cdot \left( \frac{p_j}{p_w} \right)^i \cdot \left( \frac{p_w - p_j}{p_w} \right)^{m-i} \right) \quad (8)$$

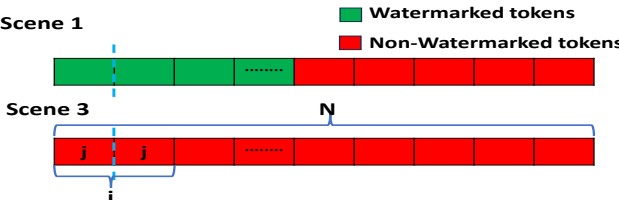

Figure 9: Assume that $\frac{N}{L} = 1$. When token $j$ does not belong to the watermark vocabulary, there are two scenes of token distribution.

Then we need to analyze two scenes where the token $j$ does not belong to the watermark vocabulary, as shown in Figure 9.

When token $j$ does not belong to the watermark vocabulary, Scene 1 appears and the probability of outputting $j$ is 0.

When token $j$ does not belong to the watermark vocabulary, Scene 3 appears, $m = 0$ and the watermark subspace is randomly filled by no-watermarked token. The probability of selecting $j$ from $N$ non-watermarked tokens is $\frac{i}{N}$, the probability of outputting $j$ is:

$$(1 - p_w)^N \cdot \sum_{i=1}^{N-m} \frac{i}{N} \cdot \binom{N-m}{i} \cdot \left( \frac{p_j}{1 - p_w} \right)^i \cdot \left( \frac{1 - p_w - p_j}{1 - p_w} \right)^{N-m-i} \quad (9)$$

The expected probability of outputting token $j$ is calculated as:

$$\sum_{m=1}^{N} \left( \frac{i}{m} \cdot \left[ \binom{N}{m} \cdot p_w^m \cdot (1-p_w)^{N-m} \right] \cdot \sum_{i=1}^{m} \binom{m}{i} \cdot \left( \frac{p_j}{p_w} \right)^i \cdot \left( \frac{p_w - p_j}{p_w} \right)^{m-i} \right)$$
$$+ (1-p_w)^N \cdot \sum_{i=1}^{N-m} \frac{i}{N} \cdot \binom{N-m}{i} \cdot \left( \frac{p_j}{1-p_w} \right)^i \cdot \left( \frac{1-p_w - p_j}{1-p_w} \right)^{N-m-i} \tag{10}$$

Let $G(z) = \sum_{i=0}^{m} z^i \cdot \binom{m}{i} \cdot p_1^i \cdot (p_2 - p_1)^{m-i} = (p_1 \cdot z + (p_2 - p_1))^m$. Taking the derivative and setting $z = 1$, we get: $G'(1) = m \cdot p_1 \cdot (p_2)^{m-1}$

As a result,

$$\sum_{i=0}^{m} i \cdot \binom{m}{i} \cdot p_1^i \cdot (p_2 - p_1)^{m-i} = m \cdot p_1 \cdot (p_2)^{m-1} \tag{11}$$

Let $p_1 = \frac{p_j}{p_w}$ and $p_2 - p1 = \frac{p_w - p_j}{p_w}$.

$$\sum_{i=1}^{m} i \cdot \binom{m}{i} \cdot \left( \frac{p_j}{p_w} \right)^i \cdot \left( \frac{p_w - p_j}{p_w} \right)^{m-i} = m \cdot \frac{p_j}{p_w} \cdot 1^{m-1} \tag{12}$$

Similarly,

$$\sum_{i=0}^{N-m} i \cdot \binom{N-m}{i} \cdot \left( \frac{p_j}{1-p_w} \right)^i \cdot \left( \frac{1-p_w - p_j}{1-p_w} \right)^{N-m-i} = (N-m) \cdot \frac{p_j}{1-p_w} \cdot 1^{N-m-1} \tag{13}$$

Substituting these results back, we get the expected probability of outputting token $j$ as:

$$\frac{1}{N} \cdot \sum_{m=1}^{N} \left( \frac{1}{m} \cdot \binom{N}{m} \cdot p_w^m \cdot (1-p_w)^{N-m} \cdot m \cdot \frac{p_j}{p_w} \right) + \left( 1 - \frac{1}{N} \right) \cdot \frac{p_j}{1-p_w} \cdot (1-p_w)^N \tag{14}$$

It can be reduced to the following.

$$\frac{p_j}{N} \cdot \sum_{m=1}^{N} \binom{N}{m} \cdot p_w^{m-1} \cdot (1-p_w)^{N-m} + \left( 1 - \frac{1}{N} \right) \cdot p_j \cdot (1-p_w)^{N-1} \tag{15}$$

Considering that the selection of the watermark vocabulary is completely random, we have $p_w = \sum_{i=1}^{L} p_i \cdot \frac{1}{N} = \frac{1}{N}$. Substituting this, we obtain the following:

$$p_j \cdot \left( \sum_{m=0}^{N} \left[ \binom{N}{m} \cdot p_w^m \cdot (1-p_w)^{N-m} \right] - (1-p_w)^N \right) + p_j \cdot (1-p_w)^N = p_j$$

The expected probability of outputting token $j$ after embedding the watermark is also consistent with that without embedding (both $p_j$) in this setting. This result demonstrates that the SPO method remains unbiased under conditions $N/L = 1$, regardless of $N$ and $L$.

Finally, we try to prove that the SPO method is unbiased in any reasonable hyperparameter settings. Assuming that $\frac{N}{L}$ is an integer greater than 1, we analyze the excepted probability of outputting token $j$. The output tokens are made up of $i$ token $j$ ($0 \leq i \leq N$) and $m$ watermarked tokens ($0 \leq m \leq N$).

We analyze the situation that token $j$ belongs to the watermark vocabulary, which contains three scenes as illustrated in Figure 10.

Figure 10: When token $j$ belongs to the watermark vocabulary, there are three scenes of token distribution.

When token $j$ belongs to the watermark vocabulary, Scene 1 appears, $m \geq \frac{N}{L}$ and the watermark subspace is randomly filled by watermarked tokens. The probability of selecting and outputting $j$ from $m$ watermarked tokens is $\frac{i}{m}$. The overall probability of outputting $j$ is given by:

$$\sum_{m=\frac{N}{L}}^{N} \left( \frac{1}{m} \cdot \binom{N}{m} \cdot p_w^m \cdot (1-p_w)^{N-m} \cdot \sum_{i=1}^{m} i \cdot \binom{m}{i} \cdot \left(\frac{p_j}{p_w}\right)^i \cdot \left(\frac{p_w - p_j}{p_w}\right)^{m-i} \right) \quad (16)$$

When token $j$ belongs to the watermark vocabulary, Scene 2 appears, $m < \frac{N}{L}$, the watermark subspace is randomly filled by $m$ watermarked tokens and $\frac{N}{L} - m$ non-watermarked tokens. The probability of selecting watermarked tokens is $\frac{m}{\frac{N}{L}}$, and the probability of selecting $j$ among them remains $\frac{i}{m}$. Thus, the probability of outputting $j$ is:

$$\sum_{m=1}^{\frac{N}{L}-1} \left( \frac{1}{\frac{N}{L}} \cdot \binom{N}{m} \cdot p_w^m \cdot (1-p_w)^{N-m} \cdot \sum_{i=1}^{m} i \cdot \binom{m}{i} \cdot \frac{p_j}{p_w}^i \cdot \left(\frac{p_w - p_j}{p_w}\right)^{m-i} \right) \quad (17)$$

When token $j$ belongs to the watermark vocabulary, Scene 3 appears, $m = 0$ and the probability of outputting $j$ is 0.

So, when token $j$ belongs to the watermark vocabulary, the expected probability of outputting $j$ is:

$$\sum_{m=\frac{N}{L}}^{N} \left( \frac{1}{m} \cdot \binom{N}{m} \cdot p_w^m \cdot (1-p_w)^{N-m} \cdot \sum_{i=1}^{m} i \cdot \binom{m}{i} \cdot \left(\frac{p_j}{p_w}\right)^i \cdot \left(\frac{p_w - p_j}{p_w}\right)^{m-i} \right) +$$
$$\sum_{m=1}^{\frac{N}{L}-1} \left( \frac{1}{\frac{N}{L}} \cdot \binom{N}{m} \cdot p_w^m \cdot (1-p_w)^{N-m} \cdot \sum_{i=1}^{m} i \cdot \binom{m}{i} \cdot \left(\frac{p_j}{p_w}\right)^i \cdot \left(\frac{p_w - p_j}{p_w}\right)^{m-i} \right) \quad (18)$$

Setting $p_1 = \frac{p_j}{p_w}$ and $p_2 - p1 = \frac{p_w - p_j}{p_w}$, we have:

$$\sum_{i=1}^{m} i \cdot \binom{m}{i} \cdot \left(\frac{p_j}{p_w}\right)^i \cdot \left(\frac{p_w - p_j}{p_w}\right)^{m-i} = m \cdot \frac{p_j}{p_w} \cdot 1^{m-1} \quad (19)$$

Substituting these results back, the expected probability of outputting token $j$ becomes:

$$\frac{p_j}{p_w} \cdot \sum_{m=\frac{N}{L}}^{N} \binom{N}{m} \cdot p_w^m \cdot (1-p_w)^{N-m} + \frac{p_j}{p_w} \cdot \frac{L}{N} \cdot \sum_{m=1}^{\frac{N}{L}-1} m \cdot \binom{N}{m} \cdot p_w^m \cdot (1-p_w)^{N-m} \quad (20)$$

Simplifying further:

$$\frac{p_j}{p_w} \cdot \left( \sum_{m=\frac{N}{L}}^{N} \binom{N}{m} \cdot p_w^m \cdot (1-p_w)^{N-m} + \frac{L}{N} \cdot \sum_{m=1}^{\frac{N}{L}-1} m \cdot \binom{N}{m} \cdot p_w^m \cdot (1-p_w)^{N-m} \right) \quad (21)$$

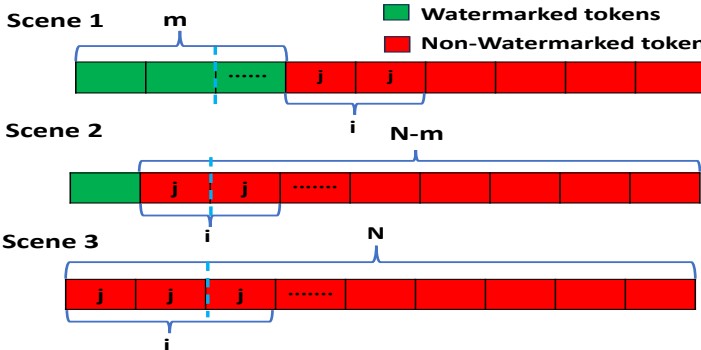

Figure 11: When token $j$ does not belong to the watermark vocabulary, there are three scenes of token distribution.

Then come to the situation that token $j$ does not belong to the watermark vocabulary with three scenes in Figure 11.

When token $j$ does not belong to the watermark vocabulary, Scene 1 appears and the probability of outputting $j$ is 0.

When token $j$ does not belong to the watermark vocabulary, Scene 2 appears, $m < \frac{N}{L}$ and the watermark subspace is randomly filled by $m$ watermarked tokens and $\frac{N}{L} - m$ non-watermarked tokens. The probability of selecting non-watermarked tokens is $\frac{\frac{N}{L}-m}{\frac{N}{L}}$, and the probability of selecting $j$ among them remains $\frac{i}{N-M}$. The probability of outputting $j$ is given by

$$\sum_{m=1}^{N/L-1} \left( \frac{N/L - m}{N/L} \cdot \binom{N}{m} \cdot p_w^m \cdot (1-p_w)^{N-m} \cdot \sum_{i=1}^{N-m} \frac{i}{N-m} \cdot \binom{N-m}{i} \cdot \left( \frac{p_j}{1-p_w} \right)^i \cdot \right.$$
$$\left. \left( \frac{1-p_w-p_j}{1-p_w} \right)^{N-m-i} \right)$$

$$(22)$$

When token $j$ does not belong to the watermark vocabulary, Scene 3 appears, $m = 0$ and the watermark subspace is randomly filled by $\frac{N}{L}$ non-watermarked tokens. The probability of selecting $j$ from $N$ non-watermarked tokens is $\frac{i}{N}$. The probability of outputting $j$ is

$$(1-p_w)^N \cdot \frac{1}{N} \cdot \sum_{i=1}^{N} i \cdot \binom{N}{i} \cdot \left( \frac{p_j}{1-p_w} \right)^i \cdot \left( \frac{1-p_w-p_j}{1-p_w} \right)^{N-i} \quad (23)$$

The preceding equation may be rewritten as

$$\sum_{m=0}^{0} \left( \frac{N/L-m}{N/L} \cdot \frac{1}{N-m} \cdot \binom{N}{m} \cdot p_w^m \cdot (1-p_w)^{N-m} \cdot \right.$$
$$\left. \sum_{i=1}^{N-m} \left[ i \cdot \binom{N-m}{i} \left( \frac{p_j}{1-p_w} \right)^i \left( \frac{1-p_w-p_j}{1-p_w} \right)^{N-m-i} \right] \right)$$

$$(24)$$

When token $j$ does not belong to the watermark vocabulary, the expected probability of outputting token $j$ is calculated as

$$
\sum_{m=1}^{N/L-1} \left( \frac{N/L-m}{N/L} \cdot \binom{N}{m} \cdot p_w^m \cdot (1-p_w)^{N-m} \cdot \sum_{i=1}^{N-m} \frac{i}{N-m} \cdot \binom{N-m}{i} \cdot \left( \frac{p_j}{1-p_w} \right)^i \cdot \right.
$$
$$
\left. \left( \frac{1-p_w-p_j}{1-p_w} \right)^{N-m-i} \right) \quad + \quad \sum_{m=0}^{0} \left( \frac{N/L-m}{N/L} \cdot \frac{1}{N-m} \cdot \binom{N}{m} \cdot p_w^m \cdot (1-p_w)^{N-m} \cdot \right.
$$
$$
\left. \sum_{i=1}^{N-m} \left[ i \cdot \binom{N-m}{i} \cdot \left( \frac{p_j}{1-p_w} \right)^i \cdot \left( \frac{1-p_w-p_j}{1-p_w} \right)^{N-m-i} \right] \right)
$$
(25)

It can simplify to

$$
\sum_{m=0}^{N/L-1} \left( \frac{N/L-m}{N/L} \cdot \frac{1}{N-m} \cdot \binom{N}{m} \cdot p_w^m \cdot (1-p_w)^{N-m} \cdot \right.
$$
$$
\left. \sum_{i=1}^{N-m} \left[ i \cdot \binom{N-m}{i} \cdot \left( \frac{p_j}{1-p_w} \right)^i \cdot \left( \frac{1-p_w-p_j}{1-p_w} \right)^{N-m-i} \right] \right)
$$
(26)

We similarly derive $\sum_{i=0}^{N-m} \binom{N-m}{i} \left( \frac{p_j}{1-p_w} \right)^i \cdot \left( \frac{1-p_w-p_j}{1-p_w} \right)^{N-m-i} \cdot i = (N-m) \cdot \frac{p_j}{1-p_w} \cdot 1^{N-m-1}$

Substituting this result, we obtain

$$
\frac{p_j}{1-p_w} \cdot \sum_{m=0}^{N/L-1} \frac{N/L-m}{N/L} \cdot \binom{N}{m} \cdot p_w^m \cdot (1-p_w)^{N-m}
$$
(27)

In summary, the expected probability of outputting token $j$ is calculated as

$$
\frac{1}{N} \cdot \frac{p_j}{p_w} \cdot \sum_{m=N/L}^{N} \binom{N}{m} \cdot p_w^m \cdot (1-p_w)^{N-m} +
$$
$$
\frac{1}{N} \cdot \frac{p_j}{p_w} \cdot \frac{L}{N} \sum_{m=1}^{N/L-1} m \cdot \binom{N}{m} \cdot p_w^m \cdot (1-p_w)^{N-m} +
$$
$$
\left( 1 - \frac{1}{N} \right) \cdot \frac{p_j}{1-p_w} \cdot \sum_{m=0}^{N/L-1} \frac{N/L-m}{N/L} \cdot \binom{N}{m} \cdot p_w^m \cdot (1-p_w)^{N-m}
$$
(28)

Considering that the watermark vocabulary is chosen completely at random, $p_w$, the probability of outputting a watermarked token is calculated by $\sum_{i=1}^{L} p_i \cdot \frac{1}{N} = \frac{1}{N}$, the equation simplifies to

$$
p_j \cdot \sum_{m=N/L}^{N} \binom{N}{m} \cdot \left( \frac{1}{N} \right)^m \cdot \left( 1 - \frac{1}{N} \right)^{N-m} +
$$
$$
p_j \cdot \frac{L}{N} \cdot \sum_{m=N/L}^{N} m \cdot \binom{N}{m} \cdot \left( \frac{1}{N} \right)^m \cdot \left( 1 - \frac{1}{N} \right)^{N-m} +
$$
$$
p_j \cdot \sum_{m=0}^{N/L-1} \frac{N/L-m}{N/L} \cdot \binom{N}{m} \cdot \left( \frac{1}{N} \right)^m \cdot \left( 1 - \frac{1}{N} \right)^{N-m}
$$
(29)

This further simplifies to

$$p_j \cdot \left[ \sum_{m=N/L}^{N} \binom{N}{m} \cdot \left(\frac{1}{N}\right)^m \cdot \left(1 - \frac{1}{N}\right)^{N-m} + \sum_{m=0}^{N/L-1} \binom{N}{m} \cdot \left(\frac{1}{N}\right)^m \cdot \left(1 - \frac{1}{N}\right)^{N-m} \right] +$$

$$p_j \cdot \frac{L}{N} \cdot \left[ \sum_{m=1}^{N/L-1} m \cdot \binom{N}{m} \cdot p_w^m \cdot (1 - p_w)^{N-m} - \sum_{m=0}^{N/L-1} m \cdot \binom{N}{m} \cdot \left(\frac{1}{N}\right)^m \cdot \left(1 - \frac{1}{N}\right)^{N-m} \right] \tag{30}$$

The combined terms within the first set of brackets form a collectively exhaustive events with a value of 1, while the terms in the second set of brackets cancel each other out, yielding a value of 0. When embedding the watermark, the expected probability of outputting token $j$ is

$$p_j \cdot 1 + p_j \cdot 0 = p_j$$

The above theoretical analysis demonstrates that regardless of the hyperparameters $N$ and $L$, the expected probability of outputting token $j$ remains consistent before and after the embedding of the watermark (both $p_j$). This fully corroborates the unbiased nature of the proposed SPO method. Although existing biased watermarking methods exhibit superior detectability and robustness, they might greatly degrade the quality of LLM's output, leading to a decrease in users' satisfaction and limiting widespread adoption of this kind of watermarking method.

## B  APPENDIX OF ALGORITHM

### B.1  IMPROVED EMBEDDING ALGORITHM FOR SPO METHOD

Despite that the SPO method achieves black-box embedding, unbiased output, and strong robustness, with overall performance exceeding typical watermarking methods, it still has certain limitations: the relatively time-consuming embedding process. The SPO method effectively addresses the issue of watermark embedding by prioritizing the allocation of watermarked tokens. However, the efficiency of this method remains less than ideal, mainly due to the following two factors: prior to each embedding operation, the SPO method requires two rounds of traversal over candidate tokens to generate watermark subspace. The efficiency of this process is greatly influenced by the size of the vocabulary and the number of candidate tokens, $N$. It is negligible for the growth of time consumption for models with smaller vocabulary (e.g., an OPT model with 50,267 tokens), but the process becomes more time-consuming for models with larger vocabularies, leading to a certain degree of reduction in output efficiency.

To optimize the watermark embedding process, we propose the following two cases based on the characteristic that only the watermark subspace is selected before final output:

Case 1: When there are empty positions in the watermark subspace after the first round of allocation (Step 1 in Figure 2), a second round of allocation (Step 2 in Figure 2) is required. In this case, tokens are dequeued from the queue and inserted into the empty positions of the subspace in sequence until the watermark subspace is filled. Once filled up, the dequeuing process can be terminated and turns to the process of outputting token immediately.

Case 2: When there are no empty positions in the watermark subspace, even if a second round allocation is performed, it will not affect the final output result. Therefore, once the watermark subspace is filled, the allocation process can be terminated, then the process of outputting token is conducted directly.

In both cases, it is unnecessary to traverse the entire candidate tokens to obtain the required subspace. Therefore, we implement relevant optimizations in the Algorithm 2. Specifically, we move the selection of the watermark vocabulary and its corresponding subspace before the traversal of the candidate tokens to facilitate the early termination of the loop. During the first round of allocation, we check if the watermark subspace is full; if so, we perceive that Case 1 occurs and terminate the loop. Similarly, during the second round of allocation, we check if the watermark subspace is full; if so, we perceive that Case 2 occurs and terminate the loop. After that, sample the output token from the watermark subspace directly to finish the embedding process.

---

**Algorithm 2** Improved SPO Embedding Method

---

**Input**: LLM $M$, prompt $p$, hyperparameter $N$ and $L$, watermark key $k$, vocabulary $V$
**Output**: generated token $x$

   Obtain token list $T$ containing $N$ candidate tokens from LLM $M$ according to prompt $p$
   Partition L subvocabularies from vocabulary $V$
   Construct subspace list $S$ with $L$ subspaces. Each subspace contains N/L positions
   Select subvocabulary $V[w]$ as watermark list according to $k$
   **for** $j = 0, 1, \ldots, N-1$ **do**
      **if** $S[w]$ has no empty position **then**
         break
      **end if**
      Get the index $i$ of token $T[j]$ according to subvocabularies
      **if** $S[i]$ has empty position **then**
         Put $T[j]$ in $S[i]$ sequently
      **else**
         Put $T[j]$ in queue
      **end if**
   **end for**
   **for** $j = 0, 1, \ldots, L-1$ **do**
      **if** $S[w]$ has no empty position **then**
         break
      **end if**
      **if** $S[j]$ has empty position **then**
         Dequeue and put in empty position of $S[j]$ subsequently
      **end if**
   **end for**
   Sample output token $x$ randomly from corresponding subspace $S[w]$
   **return** $x$

---

These improvements significantly enhance the efficiency of the watermark embedding process, reducing the expected embedding time by approximately 50% compared to the original SPO methods, thereby improving the overall performance of the watermarking algorithm. However, these optimizations do not alter the inherent time complexity, which is fundamentally determined by the nature of the watermark embedding algorithm. Future research will focus on developing more efficient and straightforward embedding methods to further optimize the prioritization process.

## B.2   DETECTION ALGORITHM FOR SPO METHOD

---

**Algorithm 3** SPO Detection Method

---

**Input**: text $x_{0:n}$, watermark key $k$, vocabulary $V$, threshold $Z_\alpha$

1: Initialize $W=0$
2: **for** Index $i = 1, 2, \ldots, n$ **do**
3:    Select watermark vocabulary $V[w]$ according to $k$
4:    **if** $x_i \in V[w]$ **then**
5:       $W++$
6:    **end if**
7: **end for**
8: Calculate Z-score according to $W$ and $L$ by Equation 1
9: **if** Z-score $<$ threshold $Z_\alpha$ **then**
10:    Report no watermark
11: **else**
12:    Report Watermarked
13: **end if**

---

## C APPENDIX OF EXTRA EXPERIMENTS

### C.1 SETTINGS AND DATASETS

**Roubustness** We gather 1,000 questions from the C4 (Raffel et al., 2020) dataset and use them as prompts. Then we choose the OPT-6.7B (Zhang et al., 2022) model and set the sampling method as top-K=50 to generate datasets. The maximum generation length for each text is set to 50, 100, and 200 respectively. We use a random perturbation parameter $\epsilon$ to create datasets subjected to different attack strengths and choose three types of typical attacks: addition, deletion, and replacement. For example, the addition dataset with $\epsilon = 0.1$ means 10% of tokens is under malicious attack.

**Unbiasedness** We follow the same evaluation process (Hu et al., 2024) to show the unbiased feature of our method. We evaluate the performance of SPO with two seq2seq models: machine translation (MT) and text summarization (TS). For the TS task, our experiment employs the BART-large model (Liu et al., 2020) as the generator, and the CNN-DM corpus (Hermann et al., 2015) as the test dataset. The MT task focuses on English-to-Romanian translation and employs the Multilingual BART (MBart) model (Liu et al., 2020) on the WMT'14 En-Ro corpus dataset.

**Detectability** We get 1,000 questions from the C4 (Raffel et al., 2020) dataset and use them as prompts. Then we use the OPT-6.7B (Zhang et al., 2022) model and set the sampling method to top-K=50 to generate datasets. The maximum generation length for each text is set to 50, 100, and 200 respectively.

**Applicability** We get 1,000 questions from the C4 dataset (Raffel et al., 2020) and PubMedQA (Jin et al., 2019) dataset. Then we use the OPT-6.7B (Zhang et al., 2022) and deepseek-llm-7b-base (DeepSeek-AI, 2024) model and set the sampling method to top-K=50 to generate datasets. The maximum generation length for each text is set to 100.

**Hyperparameter study** We prepare 1,000 questions from the C4 (Raffel et al., 2020) dataset. Then we use the OPT-6.7B (Zhang et al., 2022) and set the sampling method to top-K=50 to generate datasets. The maximum generation length for each text is set to 100 and the perturbation parameter $\epsilon$ of replacement attack is 0.1.

### C.2 BASELINE AND EVALUATION METRICS

**Robustness** We use four typical watermark methods. For biased KGW (Kirchenbauer et al., 2023a), we set KGW with $\gamma = 0.5$ and $\delta$ in $\{1.0, 2.0\}$. Meanwhile, we use two reweighting methods (Hu et al., 2024), DiPmark (Wu et al., 2024) with partition parameter $\alpha = 0.3$ and STA-1 (Mao et al., 2025) as typical unbiased methods. We report AUC in different attack settings to assess the robustness of the watermarking method.

**Unbiasedness** We choose three different watermark methods. For KGW (Kirchenbauer et al., 2023a), we set hyperparameter $\gamma = 0.5$ and $\delta$ in $\{1.0, 2.0\}$. Meanwhile, we use two reweighting methods (Hu et al., 2024) and another method DiPmark (Wu et al., 2024) with partition parameter $\alpha = 0.3$ as typical unbiased methods. We use BLEU, BERTSCORE, ROUGLE-1, and Perplexity to assess the performance of the watermarking method in different generation tasks.

**Detectability** We use four typical watermark methods. For biased KGW (Kirchenbauer et al., 2023a) , we set KGW with $\gamma = 0.5$ and $\delta$ in $\{1.0, 2.0\}$. Then we set STA-M (Mao et al., 2025) with hyperparameter M in 8, 16 as other biased method. Meanwhile, we use two reweighting methods (Hu et al., 2024), DiPmark (Wu et al., 2024) with partition parameter $\alpha = 0.3$ and STA-M (Mao et al., 2025) with hyperparameter M=1 as typical unbiased methods. We report TPR (True Negative Rate) under fixed FPR to assess the detectability of the watermarking method.

**Applicability** Similar to the detectability study, we report TPR (True Negative Rate) under fixed FPR to assess the applicability of the watermarking method on different datasets and models.

### C.3 ADDITIONAL EXPERIMENTS OF ROBUSTNESS

From the perspective of practical application, large-scale modification removes watermarks but could significantly impact the quality of the text. If the text quality deteriorates substantially after tampering, even if the attacker successfully removes the watermark, their intended objective remains

Table 4: Additional experiments of robustness: AUC of watermark detection for different methods under different perturbation strength of addition attack and max-length of generation

| Method | strength=0.0 | | | strength=0.05 | | | strength= 0.1 | | | strength=0.2 | | |
|---|---|---|---|---|---|---|---|---|---|---|---|---|
| | 50 | 100 | 200 | 50 | 100 | 200 | 50 | 100 | 200 | 50 | 100 | 200 |
| KGW ($\delta$=1) | 0.914 | 0.945 | 0.946 | 0.903 | 0.935 | 0.939 | 0.889 | 0.929 | 0.930 | 0.868 | 0.907 | 0.918 |
| KGW ($\delta$=2) | 0.980 | 0.983 | 0.983 | 0.976 | 0.980 | 0.983 | 0.971 | 0.978 | 0.983 | 0.958 | 0.970 | 0.997 |
| STA-1 | 0.991 | 0.991 | 0.989 | 0.989 | 0.989 | 0.985 | 0.987 | 0.985 | 0.983 | 0.981 | 0.978 | 0.975 |
| $\delta$-reweight | 0.864 | 0.916 | 0.932 | 0.807 | 0.858 | 0.899 | 0.720 | 0.760 | 0.809 | 0.611 | 0.653 | 0.659 |
| $\gamma$-reweight | 0.966 | 0.971 | 0.981 | 0.953 | 0.960 | 0.975 | 0.885 | 0.929 | 0.957 | 0.748 | 0.822 | 0.884 |
| DiPmark ($\alpha$=0.3) | 0.946 | 0.956 | 0.962 | 0.910 | 0.932 | 0.947 | 0.833 | 0.888 | 0.913 | 0.727 | 0.872 | 0.831 |
| SPO (N=20,L=20) | 0.996 | 0.992 | 0.993 | 0.993 | 0.992 | 0.993 | 0.991 | 0.992 | 0.991 | 0.990 | 0.991 | 0.990 |

Table 5: Additional experiments of robustness: AUC of watermark detection for different methods under different perturbation strength of deletion attack and max-length of generation

| Method | strength=0.0 | | | strength=0.05 | | | strength=0.1 | | | strength=0.2 | | |
|---|---|---|---|---|---|---|---|---|---|---|---|---|
| | 50 | 100 | 200 | 50 | 100 | 200 | 50 | 100 | 200 | 50 | 100 | 200 |
| KGW ($\delta$=1) | 0.914 | 0.945 | 0.946 | 0.904 | 0.939 | 0.940 | 0.893 | 0.926 | 0.935 | 0.886 | 0.907 | 0.919 |
| KGW ($\delta$=2) | 0.980 | 0.983 | 0.983 | 0.978 | 0.980 | 0.983 | 0.972 | 0.997 | 0.982 | 0.961 | 0.970 | 0.976 |
| STA-1 | 0.991 | 0.991 | 0.989 | 0.990 | 0.988 | 0.986 | 0.988 | 0.986 | 0.984 | 0.980 | 0.977 | 0.977 |
| $\delta$-reweight | 0.864 | 0.916 | 0.932 | 0.803 | 0855 | 0.898 | 0.716 | 0.795 | 0.816 | 0.623 | 0.666 | 0.680 |
| $\gamma$-reweight | 0.966 | 0.971 | 0.981 | 0.948 | 0.960 | 0.974 | 0.874 | 0.921 | 0.950 | 0.717 | 0.797 | 0.859 |
| DiPmark ($\alpha$=0.3) | 0.946 | 0.956 | 0.962 | 0.906 | 0.932 | 0.945 | 0.834 | 0.886 | 0.915 | 0.701 | 0.779 | 0.805 |
| SPO (N=20,L=20) | 0.996 | 0.992 | 0.993 | 0.993 | 0.992 | 0.993 | 0.993 | 0.991 | 0.992 | 0.989 | 0.990 | 0.989 |

challenging to achieve. From the standpoint of model usage, the ownership of model-generated content, particularly modified text (e.g., AI-assisted creation), remains uncertain. Therefore, we restrict the perturbation parameter $\epsilon$ in 0.05, 0.1, 0.2, ensuring that the impact on the text remains within an acceptable range and employ multiple methods for comparison.

To evaluate the robustness of the SPO method, we conduct additional tests on the same dataset. Table 4 and 5 present the AUC results when faced with common addition and deletion attacks, respectively, alongside comparisons with the mainstream unbiased and biased methods. The results demonstrate that the SPO method exhibits better robustness under attack, with only a slight decrease in AUC value. In contrast, the unbiased $\delta$-reweight (Hu et al., 2024) method shows relatively weak robustness, while DiPmark (Wu et al., 2024) and $\gamma$-reweight (Hu et al., 2024) methods, although improved robustness, still fall short of the KGW (Kirchenbauer et al., 2023a) method. Theoretically, different generation lengths affect the robustness of watermark, determined by the Z-test detection. However, in datasets with generation lengths of 50, 100, 200, the robustness of the SPO method is not significantly compromised. This outcome stems from the approach of prioritizing the allocation to maximize the number of watermarked tokens in watermark subspace, thereby increasing the success rate of the watermark embedding. Compared to other methods, the SPO method consistently maintains a high AUC by requiring more modified tokens to remove watermarks, effectively fulfilling its watermark function even under attack.

### C.4 ADDITIONAL EXPERIMENTS OF DETECTABILITY

To demonstrate the superior performance of the SPO method, we conduct additional experiments to evaluate its detectability. We set max-length of generation as 50 and 200, for these settings are commonly encountered in practical applications, and employ the same models and datasets used in the former experiments. Table 6 and 7 present the detectability results of the SPO method and provide comparisons with the mainstream watermarking methods. The experimental results indicate that the SPO method maintains excellent detectability in different length settings, and variations in the model and dataset have no significant impact on the performance of the SPO watermark. This suggests that the SPO method can effectively embed the watermark, fulfilling the requirements of practical applications.

Furthermore, Table 6 and 7 illustrate the average $p$-value of the watermarked dataset to investigate the relationship between $p$-value and detectability. The results reveal that the SPO method exhibits

Table 6: Additional experiments of detectability: TPR of generated-text detection for different methods under fixed FPR when max-length of generation is 50

| Type | Method | TPR@FPR=0.1 | TPR@FPR=0.05 | TPR@FPR=0.01 | Average $p$-value |
|------|--------|-------------|--------------|--------------|-------------------|
| Biased | KGW ($\delta$=1) | 0.762 | 0.686 | 0.412 | $1.23 \times 10^{-1}$ |
| | KGW ($\delta$=2) | 0.946 | 0.924 | 0.837 | $3.02 \times 10^{-2}$ |
| | STA-8 | 0.999 | 0.999 | 0.999 | $2.47 \times 10^{-4}$ |
| | STA-16 | 1.000 | 1.000 | 1.000 | $1.82 \times 10^{-9}$ |
| Unbiased | STA-1 | 0.983 | 0.973 | 0.837 | $7.81 \times 10^{-3}$ |
| | $\delta$-reweight | 0.747 | 0.689 | 0.864 | $4.29 \times 10^{-1}$ |
| | $\gamma$-reweight | 0.926 | 0.904 | 0.864 | $1.32 \times 10^{-1}$ |
| | DiPmark ($\alpha$=0.3) | 0.861 | 0.735 | 0.485 | $3.68 \times 10^{-2}$ |
| | SPO (N=4,L=2) | 0.912 | 0.835 | 0.593 | $3.36 \times 10^{-2}$ |
| | SPO (N=20,L=20) | 0.989 | 0.983 | 0.966 | $3.89 \times 10^{-3}$ |

Table 7: Additional experiments of detectability: TPR of generated-text detection for different methods under fixed FPR when max-length of generation is 200

| Type | Method | TPR@FPR=0.1 | TPR@FPR=0.05 | TPR@FPR=0.01 | Average $p$-value |
|------|--------|-------------|--------------|--------------|-------------------|
| Biased | KGW ($\delta$=1) | 0.861 | 0.805 | 0.540 | $5.95 \times 10^{-2}$ |
| | KGW ($\delta$=2) | 0.959 | 0.947 | 0.843 | $1.82 \times 10^{-2}$ |
| | STA-8 | 0.985 | 0.974 | 0.843 | $6.10 \times 10^{-3}$ |
| | STA-16 | 1.000 | 0.999 | 0.998 | $4.16 \times 10^{-5}$ |
| Unbiased | STA-1 | 0.987 | 0.968 | 0.821 | $4.72 \times 10^{-3}$ |
| | $\delta$-reweight | 0.897 | 0.859 | 0.809 | $2.03 \times 10^{-1}$ |
| | $\gamma$-reweight | 0.950 | 0.941 | 0.924 | $7.75 \times 10^{-2}$ |
| | DiPmark ($\alpha$=0.3) | 0.918 | 0.847 | 0.725 | $2.98 \times 10^{-2}$ |
| | SPO (N=4,L=2) | 0.925 | 0.894 | 0.779 | $2.24 \times 10^{-2}$ |
| | SPO (N=20,L=20) | 0.983 | 0.980 | 0.963 | $7.64 \times 10^{-3}$ |

a relatively low average $p$-value, indicating that under the same false positive rate (FPR) conditions, the probability of detecting watermarks from watermarked text is significantly higher, then ensure the high detectability of the watermark. Similarly to mainstream watermarking methods, the SPO method employs the Z-test to achieve detection, which involves statistically analyzing the number of tokens that meet specific conditions and conducting hypothesis testing based on the theoretical expectations established during watermark embedding. While KGW (Kirchenbauer et al., 2023a) and similar methods (Wu et al., 2024) rely on dividing tokens into red and green lists and performing hypothesis test based on the proportion of these lists, the SPO method, inspired by MCmark (Chen et al., 2025), uses the proportion of the watermark vocabulary for hypothesis testing. This approach allows the SPO method to maintain effective detection while allowing the expected proportion of watermark list detection (reflected as $\frac{1}{L}$) to be freely adjusted via the hyperparameter $L$. Importantly, this proportion directly represents the false positive rate, which reflects the probability of detecting watermarked text but reporting no watermark, thereby ensuring effective watermark embedding.

## C.5 ADDITIONAL EXPERIMENTS OF APPLICABILITY STUDY

To validate the applicability of the SPO method under various conditions, we carry out experiments and evaluate the performance of the SPO method under different settings. As shown in Table 8, 9 and 10, in two models, the SPO method consistently achieves superior performance. Specifically, for both datasets, the watermarked texts generated by the SPO method maintains a high true positive rate (TPR) under fixed false positive rates (FPR), outperforming other benchmark methods. Furthermore, the average $p$-value of the SPO method is consistently and significantly lower than that of the comparative methods. These results demonstrate the applicability of the SPO method in different models and datasets. The SPO method not only achieves a high embedding rate but also effectively performs in generated-text detection tasks, thereby validating its practical applicability in real-world scenarios.

Table 8: Additional experiments of applicability on OPT-6.7b model and PubMedQA dataset: TPR of generated-text detection for different methods under fixed FPR

| Method | TPR@FPR=0.1 | TPR@FPR=0.05 | TPR@FPR=0.01 | Average $p$-value |
|---|---|---|---|---|
| KGW ($\delta$=2) | 0.988 | 0.984 | 0.969 | $3.47 \times 10^{-3}$ |
| STA-1 | 0.993 | 0.988 | 0.963 | $3.22 \times 10^{-3}$ |
| $\delta$-reweight | 0.853 | 0.820 | 0.716 | $2.54 \times 10^{-1}$ |
| $\gamma$-reweight | 0.959 | 0.936 | 0.895 | $9.81 \times 10^{-2}$ |
| DiPmark ($\alpha$=0.3) | 0.940 | 0.919 | 0.830 | $7.33 \times 10^{-3}$ |
| SPO (N=20,L=20) | 0.995 | 0.989 | 0.986 | $1.17 \times 10^{-3}$ |

Table 9: Additional experiments of applicability on Deepseek-llm-7b-base model and C4 dataset: TPR of generated-text detection for different methods under fixed FPR

| Method | TPR@FPR=0.1 | TPR@FPR=0.05 | TPR@FPR=0.01 | Average $p$-value |
|---|---|---|---|---|
| KGW ($\delta$=2) | 0.999 | 0.995 | 0.992 | $4.83 \times 10^{-4}$ |
| STA-1 | 0.999 | 0.998 | 0.988 | $2.77 \times 10^{-4}$ |
| $\delta$-reweight | 0.243 | 0.161 | 0.056 | $9.28 \times 10^{-1}$ |
| $\gamma$-reweight | 0.808 | 0.746 | 0.581 | $3.66 \times 10^{-1}$ |
| DiPmark ($\alpha$=0.3) | 0.886 | 0.804 | 0.659 | $3.93 \times 10^{-2}$ |
| SPO (N=20,L=20) | 1.000 | 1.000 | 1.000 | $8.83 \times 10^{-19}$ |

## C.6 Additional Experiments of Unbiasedness

To investigate the impact of the hyperparameters $N$ and $L$ for the SPO method on the unbiased nature of the output, we design three additional experimental groups: when $L$=3, set $N$ as 3, 6, 9; when $N$=10, $L$=10; and when $N$=40, $L$=40. The core objective of the experiment is to compare the quality of text embedded by the SPO method with those without a watermark. As shown in Table 11, the results indicate that regardless of the hyperparameter configuration, the output quality of text with SPO watermark remains consistent with that of the non-watermarked text. This demonstrates that the watermarked text can ensure the quality of the output content. Meanwhile, this phenomenon validates that our SPO method achieves both prioritized watermarked token output and unbiasedness in generation. Such unbiased characteristics make the SPO method more suitable for practical applications, enabling content tracing without affecting the model's performance while ensuring the practicality and reliability of the watermarking technology.

## C.7 Additional Study of Hyperparameter

To systematically investigate the effects of hyperparameters $N$ and $L$ on the performance of the SPO method, we design experiments in different hyperparameter settings. Considering that both $N$ and $L$ significantly influence the generation results, we first conduct a study of the impact of $L$ on the performance of watermark with fixed $N$. We quantify the analysis by reporting the change in the AUC and average $p$-value. As shown in Figure 13, as $L$ gradually increased, the AUC of the SPO method shows a continuous upward trend, indicating that its detectability and robustness are significantly enhanced. The corresponding average $p$-value displays a progressively decreasing trend. However, when $L$ reaches 20, the average $p$-value shows a slight rebound but remains significantly lower than that of other $L$ settings (except $L = 10$). This trend suggests that reasonable enlargement $L$ within an appropriate range can effectively improve detection performance, although it is important that a larger $L$ may introduce additional trade-offs.

Next, we conduct experiments on the impact of $N$ on fixed $L$. As illustrated in Figure 12, when $N$ gradually increases, the detection performance and robustness of the SPO method demonstrate a significant improvement trend, consistent with the theoretical analysis presented in Appendix A. This indicates that increasing $N$ effectively enhances detectability and robustness.

Table 10: Additional experiments of applicability on Deepseek-llm-7b-base model and PubMedQA dataset: TPR of generated-text detection for different methods under fixed FPR

| Method | TPR@FPR=0.1 | TPR@FPR=0.05 | TPR@FPR=0.01 | Average $p$-value |
|---|---|---|---|---|
| KGW ($\delta$=2) | 0.988 | 0.983 | 0.968 | $5.18 \times 10^{-3}$ |
| STA-1 | 0.999 | 0.998 | 0.993 | $2.91 \times 10^{-4}$ |
| $\delta$-reweight | 0.164 | 0.116 | 0.035 | $9.45 \times 10^{-1}$ |
| $\gamma$-reweight | 0.491 | 0.385 | 0.184 | $7.70 \times 10^{-1}$ |
| DiPmark ($\alpha$=0.3) | 0.961 | 0.926 | 0.840 | $1.62 \times 10^{-2}$ |
| SPO (N=20,L=20) | 1.000 | 1.000 | 1.000 | $1.13 \times 10^{-33}$ |

Table 11: Additional experiments of unbiasedness: Performance of SPO method on TS and MT under different hyperparameter settings. We amplify BERTScore and ROUGE with a factor of 100

| Type | Method | Machine Translation | | Text Summarization | | |
|---|---|---|---|---|---|---|
| | | BERTScore | BLEU | BERTScore | ROUGE-1 | Perplexity |
| Baseline | No watermark | 56.2 | 21.7 | 32.67 | 38.65 | 5.031 |
| Unbiased | SPO (N=3,L=3) | 56.1 | 21.9 | 32.70 | 38.55 | 5.159 |
| | SPO (N=6,L=3) | 56.4 | 21.9 | 32.72 | 38.45 | 5.170 |
| | SPO (N=9,L=3) | 56.4 | 22.1 | 32.59 | 38.39 | 5.206 |
| | SPO (N=10,L=10) | 56.4 | 22.3 | 32.51 | 38.29 | 4.933 |
| | SPO (N=20,L=2) | 56.1 | 21.8 | 32.66 | 38.33 | 5.159 |
| | SPO (N=20,L=4) | 56.1 | 21.7 | 32.60 | 38.29 | 4.933 |
| | SPO (N=20,L=10) | 56.3 | 22.0 | 32.32 | 38.12 | 4.925 |
| | SPO (N=40,L=40) | 56.1 | 22.1 | 32.65 | 38.51 | 4.971 |

Focusing on the experimental results presented in Figure 13, we perform an exploration from the perspective of robustness and detectability. As $L$ increases, the number of subvocabularies also increases, leading to a decrease in the theoretical false positive rate. This manifestation is reflected by the decrease in average $p$-value and the enhancement of detectability. However, it is crucial to recognize that the augmentation of $L$ simultaneously results in a theoretical decrease in robustness. Specifically, under random modification, the probability of erasing a watermarked token can be calculated as $1 - \frac{1}{L}$. Consequently, an increase in $L$ inevitably increases the likelihood of a successful attack. However, the probability of erasing a single token exclusively represents one dimension of robustness, rather than being comprehensive. The SPO method employs a $Z$-score detection, where the watermark verification process depends not only on the success rate of erasing individual watermarked token but also on the number of valid watermarked tokens and their theoretical embedding success rate. This mechanism requires a multidimensional analysis of various influencing factors for robustness, rather than reliance on a single metric.

In conclusion, the configuration of the hyperparameters $N$ and $L$ significantly impacts the performance of the SPO method. Theoretically, increasing both $N$ and $L$ enhances detection performance, but practical applications still require selecting appropriate hyperparameter combinations based on specific requirements, balancing detectability and robustness.

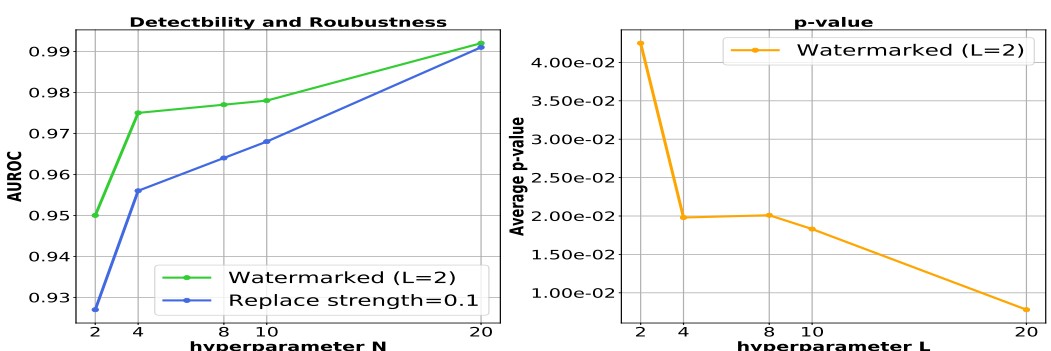

Figure 12: Additional experiments of hyperparameter when $L$=2: **Left:** AUC of watermark detection on two different dataset: watermarked dataset and watermarked dataset under replacement attack (strength=0.1), **Right:** average $p$-value of watermarked dataset.

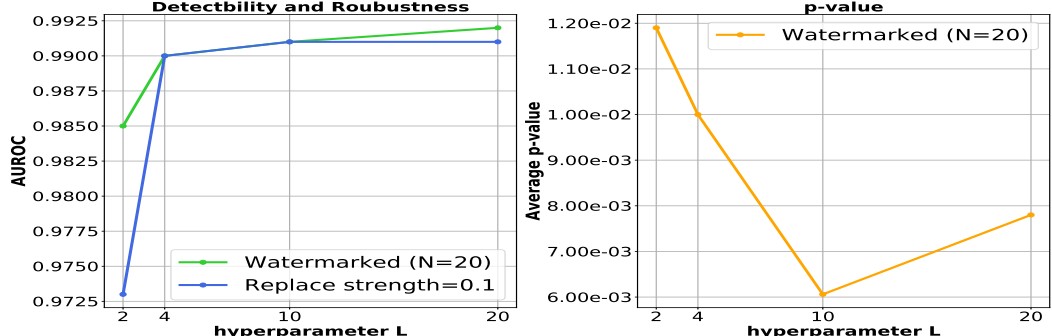

Figure 13: Additional experiments of hyperparameter when $N$=20: **Left:** AUC of watermark detection on two different dataset: watermarked dataset and watermarked dataset under replacement attack (strength=0.1), **Right:** average $p$-value of watermarked dataset.

