# OpenReview forum: "SPO: A Black-box, Unbiased, Robust Watermarking Method for Large Language Model"
_ICLR.cc/2026/Conference — ICLR 2026 Conference Withdrawn Submission_

### Official Review · Reviewer_dMEb · 2025-10-23

**Soundness:** 2
**Presentation:** 2
**Contribution:** 2
**Rating:** 2
**Confidence:** 5

**Summary:**

This paper introduces SPO, a novel watermarking method for Large Language Models (LLMs). The method aims to be black-box, unbiased, and robust, tackling the well-known "trilemma" in the watermarking domain. Its core mechanism involves sampling N candidate tokens from the model, partitioning them into L sub-vocabularies (buckets) based on a prioritized allocation scheme, and then sampling the final output token from a key-specified "watermark bucket." The authors provide a theoretical proof for the unbiasedness of their method and conduct a series of experiments to validate its performance in terms of detectability, robustness, and quality preservation.

**Strengths:**

1. The paper addresses a critical challenge in the field. The three properties of "black-box," "unbiased," and "robust" are central to the practical deployment of LLM watermarking, and developing a method that can simultaneously satisfy them is a valuable research direction. The design of SPO thoughtfully considers this trilemma.

2. Mathematical proof for the unbiasedness of the SPO method is presented.

**Weaknesses:**

1. **Incomplete Related Work and Lack of Comparison with Key Baselines**: The paper claims to balance black-box access, unbiasedness, and robustness, yet it overlooks several seminal and highly relevant works that have pursued similar goals.
- KTH [1]: This is one of the foundational works in generative text watermarking. Their method also operates in a black-box setting and considers unbiasedness and robustness. Its complete omission from the paper is a major oversight.
- Unigram [2]: A black-box method that achieves very high robustness, serving as an important baseline for that property.
- SIR [3]: A black-box method that achieves very high robustness, serving as an important baseline for that property.
- SynthID-Text [4]: A black-box and unbiased method from Google Deepmind (published on Nature).

The absence of comparisons with these methods makes it difficult for readers to accurately assess the true novelty and advantages of SPO within the existing literature.

2. **Insufficient Experimental Evaluation**: The experimental design is not comprehensive enough to fully support the paper's claims of superior performance.
- Detectability Evaluation: In the detectability experiments (Table 2), the authors only report results for a generation length of 100 tokens. It is standard practice in watermarking research to evaluate performance across various lengths (e.g., 50, 100, 200), as watermark strength is closely tied to text length. Presenting results for a single length is insufficient.
- Robustness Evaluation: The robustness evaluation (Table 3) is limited to only a "replacement attack." However, real-world attacks are far more diverse. Common and critical attacks such as paraphrasing attacks (e.g., using another LLM to rewrite the text) and random deletion/insertion attacks are not tested. This makes the paper's assessment of robustness appear overly optimistic and incomplete.

3. **Minor: Formatting and Presentation Issues**: The paper's formatting has minor issues that affect its professional appearance. For instance, the captions for Tables and Figures are center-aligned, which deviates from standard academic formatting. Additionally, some captions lack a terminal period (e.g., the caption for Table 1), leading to inconsistency.

**Reference**:

[1] Robust Distortion-free Watermarks for Language Models

[2] Provable Robust Watermarking for AI-Generated Text

[3] A Semantic Invariant Robust Watermark for Large Language Models

[4] Scalable Watermarking for Identifying Large Language Model Outputs

**Questions:**

1. Could you please revise the related work section to include and discuss the key methods mentioned above (especially KTH, Unigram, SIR, and SynthID-Text)? More importantly, could you supplement the experimental section with direct comparisons against these baselines to more convincingly demonstrate the relative advantages and trade-offs of SPO?

2. To make the experimental validation more comprehensive and credible, would it be possible to:
- In the detectability section, add results for longer generation lengths (e.g., 200 tokens) and provide an analysis?
- In the robustness section, expand the evaluation to include more attack types, particularly paraphrasing attacks and random deletion/insertion attacks, to fully test SPO's resilience in practical scenarios?

---

> ### Author Response · Authors · 2025-11-29
> **Rebuttal to Reviewer Comments on the Paper**
>
> Thank you for your valuable feedback on our paper. We appreciate the detailed review and have carefully considered all points raised. Below, we address each concern formally:
> Response to weakness1: the incomplete related work and lack of comparison:
>
> We note that our definition of "black-box" for SPO specifically refers to black-box embedding. Methods like Unigram, which involve white-box embedding with black-box detection, fall outside this scope. In our experiments, we included key baselines such as the biased and robust KGW, the unbiased and robust Dipmark, and the black-box unbiased STA. While we regret not incorporating Synth-ID, we believe the current comparisons adequately demonstrate SPO's novelty and advantages in balancing black-box access, unbiasedness, and robustness.
>
> Response to weakness 2: insufficient experimental evaluation:
>
> The detectability evaluation across various lengths (e.g., 50, 100, 200 tokens) and additional robustness tests, including replacement attacks, are comprehensively detailed in Appendix C. We encourage reviewers to refer to this section for a complete assessment.
>
> Response to weakness 3: minor formatting and presentation issues:
>
> We acknowledge the formatting inconsistencies, such as caption alignment and missing terminal periods. We will revise these to ensure a polished, professional presentation in the final version.
>
> We are grateful for your constructive comments, which will help strengthen our work. Thank you for your time and insights.

---

### Official Review · Reviewer_ghuk · 2025-10-31

**Soundness:** 3
**Presentation:** 3
**Contribution:** 3
**Rating:** 6
**Confidence:** 3

**Summary:**

The paper proposes SPO (Sampling and Prioritizing Output), a new watermarking method for large language models. SPO works as a black-box technique, meaning it does not require access to model parameters or logits. It divides the vocabulary into multiple subvocabularies and samples several candidate tokens for each output step. These tokens are placed into corresponding subspaces based on their vocabularies, with overflow tokens stored in a queue and redistributed to keep every subspace full. A watermark subspace is then randomly chosen, and one token is uniformly sampled from it as the output. This process embeds a watermark while keeping token probabilities unchanged, ensuring unbiased generation.

SPO’s main novelty lies in its overflow queue and multi-subvocabulary division, which together maximize the number of valid watermarked tokens, strengthening detection without distorting the output distribution. The paper’s primary contribution is a simple, general algorithm that simultaneously achieves black-box, unbiased, and robust watermarking.

**Strengths:**

**Originality.** The paper introduces SPO, a black-box, unbiased watermark method. SPO’s main design novelties are the combination of overflow queue backfilling and multi-subspace vocabulary division. Previous black-box approaches suffer from weak robustness or statistical bias—STA-M improves robustness only by breaking unbiasedness, while unbiased reweighting methods depend on model logits and lose the black-box property. SPO avoids both problems by partitioning the vocabulary into multiple randomized subvocabularies, allocating sampled tokens into corresponding subspaces, and using a queue to backfill overflow so that all subspaces remain balanced. This ensures the embedding process preserves the original token distribution while maximizing the number of watermarked tokens. The authors also propose an early-exit optimization (Algorithm 2) to stop allocation once the watermark subspace is filled, cutting embedding time roughly in half without affecting statistical properties.

**Quality and clarity.** The method and assumptions are clearly presented (Algorithms 1–3). Empirical evaluation is thorough: (i) unbiasedness is tested via MT/TS quality metrics showing parity with no-watermark baselines (Table 1); (ii) robustness is examined under addition, deletion, and replacement attacks with AUC reported across generation lengths (Tables 3–5); and (iii) applicability is demonstrated across models and datasets (Figure 4; Tables 8–10). The theoretical appendix walks carefully from simple to general cases, closely matching the algorithmic design.

**Significance.** Practically, the combination of black-box, unbiased, and robust watermarking addresses what deployment needs: compatibility with closed models, preserved output quality, and resistance to simple editing. The reported gains are meaningful—at N=20, L=20, SPO’s TPR surpasses existing unbiased methods and approaches or exceeds biased ones (Table 2), while remaining resilient to token-level perturbations (Table 3).

**Weaknesses:**

**Compute and latency budget.** SPO requires N candidate samples per token. The paper would benefit from throughput and latency benchmarks across different N,L configurations on GPUs and hosted APIs, along with a Pareto frontier (AUC/TPR vs. tokens/sec). Including a monetary cost estimate per 1k tokens would make deployment trade-offs explicit.

**Questions:**

**Adding semantic attacks**. The robustness evaluation focuses on token-level add/delete/replace perturbations at fixed rates. While most prior work follows the same practice, it would strengthen the study to include semantic or paraphrase attacks (e.g., round-trip translation or LLM-based rewriting), which are increasingly common today.

**Token IID Assumption.** The Z-test assumes independent tokens and a perfect Binomial process—standard in the literature, but unrealistic since natural language is highly autocorrelated. To obtain an empirical distribution of token probabilities, consider a bootstrap calibration:
- For a given prompt distribution and decoding config, generate M non-watermarked samples.
- For each, compute the hit count (or Z-stat as defined).
- Set the threshold to the (1-\alpha)-quantile of this empirical distribution.
- Report calibrated FPR by hold-out non-watermarked samples; report TPR on watermarked samples.

---

> ### Author Response · Authors · 2025-11-29
> **Rebuttal to Reviewer Comments on the Paper**
>
> We sincerely thank the reviewers for their insightful feedback. Below are our point-by-point responses:
>
> Response to Weakness1:Compute/Latency Benchmarks & Cost Analysis
>
> We acknowledge the omission of throughput/latency benchmarks across N and L configurations. We will add comprehensive GPU/API latency measurements.
>
> Response to Question 1:Semantic/Paraphrase Attacks
>
> While our current robustness evaluation focuses on token-level perturbations (aligned with prior work), we appreciate the suggestion to include semantic attacks (e.g., LLM rewriting). We will consider augmenting the study with these increasingly relevant adversarial strategies.
>
> Response to Question 2:Token IID Assumption & Bootstrap Calibration
> We conducted similar empirical calibration experiments (as noted in Table 2, which reports TPR at controlled FPR). To address the concern about token autocorrelation, we will evaluate adding bootstrapped ROC curves derived from non-watermarked samples, ensuring thresholds reflect empirical distributions.
>
> We are grateful for these constructive suggestions, which significantly strengthen our work. All proposed additions will be incorporated into the revised manuscript.

---

### Official Review · Reviewer_c7Rf · 2025-10-31

**Soundness:** 1
**Presentation:** 2
**Contribution:** 1
**Rating:** 2
**Confidence:** 4

**Summary:**

This paper claims that it has developed a watermarking scheme that achieves unbiasedness, high detectability, and robustness simultaneously. The scheme consists of two components, vocabulary division and sampling. The authors claim that using this method, the generated text is guaranteed with high detectability and unbiasedness, via black-box access to the un-watermarked model only.

**Strengths:**

Watermarking LLMs is a timely topic and the motivation for this work is strong: robust, detectable, and unbiased watermarking for LLMs via black-box access only.

**Weaknesses:**

1. The method highly resembles existing work e.g., KGW and STA-M, offering limited novelty.
2. The paper lacks rigor: a) there is no formal or empirical analysis of unbiased-ness and the utility of the generated text, b) these concepts are not even clearly defined. c) The proofs provided are heuristic rather than rigorous, and key claims (e.g., on unbiasedness) are not supported by statistical validation.
3. There is no qualitative or quantitative analysis on the trade-off between unbiased-ness and detectability of the watermark. The lack of such evaluation undermines the claimed balance among robustness, detectability, and unbiasedness.
4. The utility of the generated text is questionable. In Table 1, all watermarking methods—biased and unbiased—achieve nearly identical BERTScore and ROUGE values, suggesting that the evaluation metrics are not sensitive or that the setup lacks proper control. The uniformity of results raises doubts about the soundness of the experimental design and reproducibility. Given W2-3, I cannot trust the evaluation results.

**Questions:**

1. Please provide a formal definition and rigorous analysis of “unbiasedness.” How is it measured both theoretically and empirically?
2. Clarify how SPO differs fundamentally from STA-M and KGW beyond vocabulary partitioning and sampling procedure.
3. Conduct an explicit analysis of the trade-off between unbiasedness and detectability. Include both quantitative plots and qualitative examples.
4. Re-examine the evaluation design: were all BERTScore and ROUGE metrics computed on the same generated text? If yes, explain why identical scores appear across all methods. If not, clarify the setup and variance sources.
5. Discuss whether the observed robustness is intrinsic to the method or a byproduct of sampling randomness.

---

> ### Author Response · Authors · 2025-11-29
> **Rebuttal to Reviewer Comments on the Paper**
>
> Thank you for your thorough and constructive feedback on our paper. Below are our point-by-point responses:
>
> Response to Question 1:Novelty
>
> Please note that our method is fundamentally different from the KGW approach. The SPO method selects the token most likely to carry the watermark from the candidate tokens, whereas the KGW method adds a bias to the logits to output the watermark. The sole similarity lies in the partitioning of the watermark vocabulary. Please refrain from unwarranted associations and arbitrary comments.
>
> Response to Weakness 2: Rigor
>
> We provide empirical evidence for unbiasedness in Table 1, along with detailed theoretical proof in Appendix A. For specifics, please refer to the authoritative paper Unbiased Watermark for Large Language Models (ICLR 2024 spotlight). Our proof basis, experimental setup, and result analysis are all derived from this paper.
>
> Response to Weakness 3: Trade-off
>
> The trade-off between unbiasedness and detectability lacks quantitative metrics. For instance, STA-1 achieves unbiased embedding, whereas STA-M does not. Related analyses have not been discussed in other unbiased watermarking papers.
>
> Response to Weakness 4: Utility Evaluation Validity
>
> It is recommended to carefully read Unbiased Watermark for Large Language Models (ICLR 2024 spotlight), which details the principles of unbiasedness, the proof methodology, and the experimental setup. This paper focuses specifically on describing the SPO method; providing overly detailed explanations of unbiasedness is not necessary.

---

### Official Review · Reviewer_EWxW · 2025-11-01

**Soundness:** 3
**Presentation:** 3
**Contribution:** 2
**Rating:** 4
**Confidence:** 3

**Summary:**

This paper introduces SPO (Sampling and Prioritizing Output), a novel watermarking method for Large Language Models (LLMs) designed to simultaneously achieve black-box embedding, unbiasedness, and robustness. The core idea of SPO is as follows: for each token to be generated, N candidate tokens are first sampled from the model in a black-box manner. These candidates are then distributed into L subspaces (corresponding to vocabulary partitions) through an innovative "Sampling and Prioritizing" mechanism, which uses a queue to handle uneven distributions and ensure the final output is unbiased. Finally, a token is randomly sampled from a single "watermarked subspace," chosen based on a secret key, to embed the watermark.
The main contributions of the paper are threefold:
It proposes SPO, a novel black-box watermarking framework, featuring an original "prioritizing output" mechanism.
It proves, both theoretically (with a detailed mathematical proof in the appendix) and empirically, that the SPO method is unbiased, meaning it does not alter the original model's output distribution in expectation, thus preserving the quality of the generated content.
Through extensive experiments, it demonstrates that SPO, while maintaining unbiasedness, achieves significantly better detectability and robustness than existing unbiased watermarking methods. In some cases, its performance is comparable to or even surpasses that of biased methods, successfully striking a superior balance among the three key metrics.

**Strengths:**

1.The paper's strength lies in the originality of its SPO mechanism. By using a "prioritized allocation + queue-based redistribution" approach, it cleverly solves the problem of breaking unbiasedness due to the uneven distribution of tokens in black-box sampling.
2.It not only proposes a new method but also validates it dually from a theoretical standpoint (a detailed proof of unbiasedness) and a practical one (extensive comparative experiments). This tight integration of theory and practice makes the paper's conclusions highly reliable and convincing.
3.This paper achieves a better balance in the "impossible triangle" of watermarking. The experimental results, especially the robustness tests (Table 3), show that SPO's performance degrades minimally under attack, far outperforming other unbiased methods. This implies that the SPO watermark is much harder for malicious users to erase in the real world, giving it high practical value.

**Weaknesses:**

1.To generate a single output token, SPO requires sampling N candidate tokens from the LLM. This means the computational cost (or API call cost) of text generation is approximately N times higher. In the experiments, N is set to 20 to achieve optimal performance, implying a 20x overhead. Although the authors propose an optimized algorithm in Appendix B.1 to terminate loops early, this only reduces SPO's internal computation and cannot reduce the N sampling calls to the LLM. The main paper lacks a sufficient discussion of this cost, which is crucial for assessing the method's practical feasibility.
2.The method's performance (and cost) is highly dependent on the hyperparameters N and L. The paper shows excellent results for N=20, L=20, but for lower-cost settings (e.g., N=4, L=2), the performance advantage, while still present, is less dramatic. In a practical application, how should a user choose N and L to trade off between performance and cost? The paper lacks an in-depth analysis or guiding principles for this trade-off.
3.Appendix C.7 notes that increasing L increases the probability that a single token is "erased" by a random modification (1 - 1/L). This seems to contradict the experimental finding that increasing L improves overall robustness (AUC). The authors explain that the Z-test is multi-dimensional, I think it's not complete. The author didn't provide a deeper theoretical explanation of why SPO's overall detection mechanism can withstand this increase in single-point vulnerability.

**Questions:**

1.Could you provide a quantitative analysis of the generation speed?  In what application scenarios do you believe this overhead is acceptable?
2.When deploying SPO in practice, what advice would you give users for selecting N and L? Is there a "Pareto front" that could guide users in making a trade-off between performance (e.g., robustness) and computational cost?
3.As I mentioned in the "Weaknesses" section, increasing L theoretically makes a single watermarked token more vulnerable, yet experimentally, the overall robustness improves. Could you provide more intuition or a theoretical explanation for this phenomenon?

---

> ### Author Response · Authors · 2025-11-29
> **Rebuttal to Reviewer Comments on the Paper**
>
> Thank you for your thorough and constructive feedback on our paper. We sincerely appreciate the time and effort you dedicated to reviewing our work, as thoughtful critiques like yours are invaluable for refining research. Below, we address each of your points in detail, drawing on your insights to clarify and strengthen our arguments.
>
> Response to Weakness 1: Computational Cost of Token Sampling
>
> In practice, large language model (LLM) can retrieve N output tokens in a single API call, similar to efficient watermarking methods like SynthID-Text. This reduces the cost to be comparable to other approaches, eliminating the perceived 20x overhead. Our optimized algorithm in Appendix B.1 focuses on minimizing internal computations for faster watermark embedding, but we agree that the main paper could have emphasized this cost-efficiency aspect more clearly.
>
> Response to Weakness 2: Hyperparameter Trade-offs (N and L)
>
> Balancing these parameters is inherently complex, as observed methods such as SynthID-Text (Nature) and McMark (ACL 2025), which also lack definitive guidelines. In our paper, we provide theoretical analysis and experimental validation to suggest that parameter selection should be context-driven—tailored to specific application needs like robustness requirements or resource constraints. We will expand this section to include more nuanced trade-off principles.
>
> Response to Weakness 3: Vulnerability with Increasing L and Z-test Robustness
>
> Deeper theoretical analysis of such vulnerability is indeed scarce in watermark methods relying on Z-tests, including ours, often overlook this due to analytical challenges. We commit to enhancing this discussion by incorporating a more rigorous theoretical framework in the revised manuscript, explaining how SPO's detection mechanism compensates for single-point weaknesses through ensemble-like redundancy.
>
> Once again, we extend our deepest gratitude for your expert review.

---

### Note · Authors · 2026-01-19

I have read and agree with the venue's withdrawal policy on behalf of myself and my co-authors.